# NUAK2 is a critical YAP target in liver cancer

Wei-Chien Yuan[1,2,3], Brian Pepe-Mooney[1,2], Giorgio G. Galli[1,2,3,8], Michael T. Dill [1,2,3], Hai-Tsang Huang[4,5], Mingfeng Hao[5], Yumeng Wang[6], Han Liang [6], Raffaele A. Calogero[7] & Fernando D. Camargo[1,2,3]

The Hippo-YAP signaling pathway is a critical regulator of proliferation, apoptosis, and cell fate. The main downstream effector of this pathway, YAP, has been shown to be mis-regulated in human cancer and has emerged as an attractive target for therapeutics. A significant insufficiency in our understanding of the pathway is the identity of transcriptional targets of YAP that drive its potent growth phenotypes. Here, using liver cancer as a model, we identify NUAK2 as an essential mediator of YAP-driven hepatomegaly and tumorigenesis in vivo. By evaluating several human cancer cell lines we determine that NUAK2 is selectively required for YAP-driven growth. Mechanistically, we found that NUAK2 participates in a feedback loop to maximize YAP activity via promotion of actin polymerization and myosin activity. Additionally, pharmacological inactivation of NUAK2 suppresses YAP-dependent cancer cell proliferation and liver overgrowth. Importantly, our work here identifies a specific, potent, and actionable target for YAP-driven malignancies.

[1] Stem Cell Program, Boston Children's Hospital, Boston, MA 02115, USA. [2] Department of Stem Cell and Regenerative Biology, Harvard University, Cambridge, MA 02138, USA. [3] Harvard Stem Cell Institute, Boston, MA 02115, USA. [4] Department of Cancer Biology, Dana-Farber Cancer Institute, Boston, MA 02115, USA. [5] Department of Biological Chemistry and Molecular Pharmacology, Harvard Medical School, Boston, MA 02115, USA. [6] Department of Bioinformatics and Computational Biology, The University of Texas MD Anderson Cancer Center, Houston, TX 77030, USA. [7] Department of Biotechnology and Health Sciences, Molecular Biotechnology Center, University of Torino, Torino 10126, Italy. [8]Present address: Novartis Institutes for BioMedical Research, Disease Area Oncology, 4057 Basel, Switzerland. Correspondence and requests for materials should be addressed to F.D.C. (email: camargo@fas.harvard.edu)

Uncontrolled cell growth is a hallmark of cancer, often driven by mutations in pathways that control cell proliferation and survival. The Hippo-YAP network is one such pathway, which has also been implicated in the control of developmental transitions, organ size, regeneration, and cell fate[1–5]. The transcriptional co-activators YAP and the highly similar protein TAZ are the major downstream effectors of this pathway. YAP/TAZ are negatively regulated by a core cascade of proteins, including NF2, LATS1/2, and MST1/2. YAP/TAZ are directly phosphorylated by LATS1/2, which leads to their cytosolic retention and subsequent proteasomal degradation[6–8]. In the absence of Hippo pathway engagement, YAP/TAZ translocate into the nucleus and through interactions with the TEAD family of transcription factors, activate genetic programs involved in proliferation and survival[9,10]. Important inputs into the Hippo signaling cascade, include cell density, cell polarity, and cell tension, which signals to YAP/TAZ via the cytoskeleton[11–13].

Tight control of YAP activity is crucial for normal tissue growth and homeostasis. Experimental activation of YAP via genetic means leads to massive tissue overgrowth, stem cell expansion, and tumorigenesis[1,14]. Furthermore, YAP is required for the growth of multiple epithelial and nonepithelia tumors in mouse models[15–19]. While the frequency of mutations for components of the Hippo pathway is rare in most tumor types, a myriad of clinical evidence has shown that YAP is found overexpressed, and/or highly activated in multiple types of malignancies[20,21], and its nuclear localization is positively correlated with poor prognosis in many cancers[22–24]. Consequently, the Hippo-YAP pathway has emerged as an attractive and novel therapeutic target for oncology. However, a major caveat in developing molecules that antagonize YAP is the lack of traditional druggable molecules in the pathway. Current known kinases of the Hippo signaling pathway are growth suppressive, and therefore unsuitable as cancer targets. And while some progress has been made in developing molecules that could inhibit the YAP/TEAD interaction[25], the intrinsic nature of inhibiting protein–protein interfaces makes this approach specially challenging. Thus, the identification of traditional drug targets, i.e., enzymes, in the pathway would represent an important step forward.

Extensive work has been done to profile the genetic program regulated by YAP in multiple cell types. While several datasets have been assembled describing direct targets of YAP in various datasets, the significance of these targets to the function of YAP is unclear, especially in context of cancer. The best well studied downstream targets, for instance, (i.e., *CTGF*, *CYR61*, and *AMOTL2*) are not considered to have an oncogenic role. While other YAP-targets have been shown to have some proliferative effect in cell lines, the relevance and potency of these to suppress YAP phenotypes in vivo has not been demonstrated[9,26–29]. The identification of bona fide effectors downstream of YAP would not only provide a deeper understanding of the oncogenic mechanisms of this pathway, but could also novel actionable entry-points for YAP-driven cancer.

On the basis of the remarkable effects of YAP on liver growth and the general dependency of liver tumors on YAP activity[1,14,19], here we use the liver as a model system to elucidate important drivers of YAP function. By overlapping with the genome-wide chromatin occupancy analyses and gene expression profiling datasets, we identified the kinase NUAK2, as a critical mediator of YAP-driven growth. Furthermore, using both genetic and pharmaceutical inhibition of NUAK2, we provide evidence for its requirement in YAP-driven proliferation and tumorigenesis. Thus, NUAK2 serves as a central player in the remarkable effects of YAP on liver growth, the general dependency of liver tumors on YAP activity, and it represents an actionable entity within the pathway that could be amenable to target Hippo/YAP signaling.

## Results

**Identification of YAP target genes**. YAP overexpression in liver leads to hepatomegaly and liver tumorigenesis[1,14]. To functionally identify critical and direct YAP-targets mediating this response, we performed chromatin immunoprecipitation followed by sequencing (ChIP-Seq) for TEAD4 as a surrogate of YAP binding after inducible expression of YAP S127A in murine hepatocytes. The S127A mutation abolishes one of several phosphorylation sites of LATS1/2 and displays a partial increase in nuclear localization[30]. As shown previously in human liver cancer cell lines, we find that TEAD4 is predominantly bound to distal regulatory elements, a large fraction of which overlap with H3K27ac active enhancer marks[31] (Fig. 1a). We next assigned the corresponding target genes to the TEAD4 bound regulatory elements and assessed their relative expression in the livers of induced TetO-YAP-S127A mice (Fig. 1b). This identified bound and regulated genes were then cross-referenced to data obtained from YAP ChIP-Seq and RNA-seq experiments in HuCCT-1 human cholangiocarcinoma cells[31] to select a list of 14 direct and conserved YAP transcriptional targets (Supplementary Table 1).

***Nuak*2 is a direct target of YAP**. Considering the lack of traditional drug targets in the Hippo pathway, we decided to focus on identifying kinases, which were potentially actionable targets of pharmacological inactivation. Of the 14 targets identified only one matched this criteria, NUAK2.

NUAK2, also known as sucrose nonfermenting (SNF1)-like kinase (SNARK), belongs to the AMPK protein kinase family[32]. NUAK2 has been mostly implicated in human cancer development by amplifications in human melanoma, although copy number gains of 1q32 are found in other epithelial malignancies[33–35]. However, other data suggest that NUAK2 might have tumor suppressive roles in the context of a colorectal cancer model[36]. Thus, our understanding of the role of NUAK2 in cancer is still quite limited.

We first examined in more detail the regulation of NUAK2 expression by YAP. Analysis of ChIP-seq datasets revealed robust binding of YAP and TEAD to sequences downstream of NUAK2[31,37,38], in human cholangiocarcinoma (Fig. 1c), mesothelioma (Fig. 1d), breast adenocarcinoma, glioblastoma, and fetal lung cell lines (Supplementary Fig. 1a). Further inspection of individual peaks revealed that the YAP occupancy sites co-localized with enhancer elements, as defined by typical enhancer-associated histone posttranslational modifications (H3K27ac$^+$) (Fig. 1c, d). These enhancers were classified as super-enhancers in human cholangiocarcinoma and mesothelioma cell lines, arguing for the importance of NUAK2 in these cancers (Fig. 1f). Consistently, YAP and TEAD4 also co-occupied an enhancer downstream of mouse *Nuak2* in the liver of TetO-YAP S127A mice (Fig. 1e). Furthermore, we validated that acute YAP overexpression led to the up-regulation of *NUAK2* mRNA and protein (Fig. 1g, h). Conversely, YAP or YAP/TAZ knockdown in HuCCT-1 cells nearly abolished the expression of *NUAK2* mRNA (Fig. 1i). We also identified two putative TEAD-responsive elements (TREs) based on consensus TEAD-binding sequences in the YAP/TEAD-defined *NUAK2* enhancers (Supplementary Fig. 1b). Mutation of one of these (TRE1), but not the other, abolished YAP-driven transcriptional induction of *NUAK2* enhancer activity (Supplementary Fig. 1b). Recent work has also identified *Nuak2* as Yap transcriptional target in embryonic lungs[39]. Together, our results indicate that YAP directly regulates *NUAK2* transcription through TEAD-binding to the *NUAK2* super enhancer.

To extend our findings of YAP-mediated NUAK2 induction to human cancer, we analyzed the expression profiles of YAP

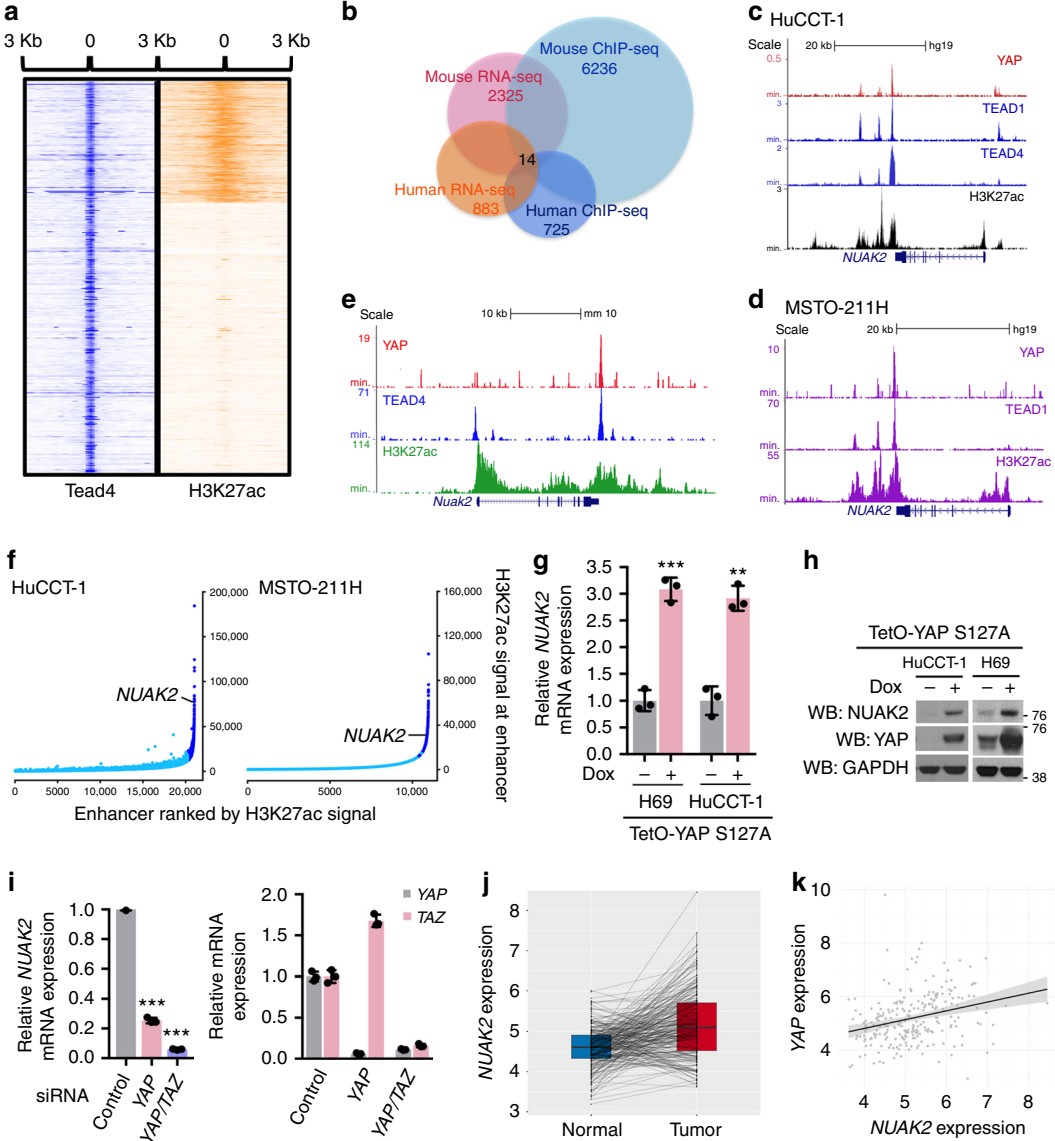

**Fig. 1** NUAK2 is a direct target of YAP. **a** Heatmap representing TEAD4/H3K27ac ChIP-seq signal in a window of ±3 Kb from the center of TEAD4 peaks. Clustering results from the K-means method. **b** Venn diagram displays overlapping genes from 4 different datasets to identify 14 accordant YAP downstream genes. The datasets include: (1) a TEAD4 ChIP-seq from the liver of induced TetO-YAP mice, (2) a YAP ChIP-seq, from human cholangiocellular carcinoma cell line, HuCCT-1, (3) an RNA-seq from HuCCT-1 cells upon *YAP/TAZ* silencing, and (4) an RNA-seq from the liver of induced TetO-YAP mice. **c, d** Genomic tracks display ChIP-seq data for the indicated antibodies around the *NUAK2* gene in HuCCT-1 (**c**) and MSTO-211H cells (**d**). **e** Genomic tracks display ChIP-seq data for the indicated antibodies around the *Nuak2* gene in primary hepatocytes of TetO-YAP S127A mice placed on Dox for 4 days. **f** Hockey-stick plot representing H3K27ac signal across enhancer regions for all enhancers in HuCCT-1 (left panel) and MSTO221H (right panel) cells. Super enhancers are labeled by dark blue, with the super enhancer of Nuak2 marked. **g** qPCR analysis of *NUAK2* expression in HuCCT-1 and H69 cells stably expressing Dox-inducible YAP-S127A. Data are presented as mean ± SD; $n = 3$. The two-tailed, Student's *t* test was used to compare between two groups and expressed as *P* values. *$P < 0.05$, **$P < 0.01$, ***$P < 0.001$. **h** YAP overexpression promotes NUAK2 expression. Dox-inducible HuCCT-1 and H69 cells treated with or without Dox for 24 h were assayed by Western blot for the indicated antibodies. **i** qPCR analysis of *NUAK2* expression in HuCCT-1 cells transfected with indicated siRNA for 72 h (left panel). Right panel showing the knockdown efficiency of *YAP* and *TAZ*. Data are presented as mean ± SD; $n = 3$. The two-tailed, Student's *t* test was used to compare between two groups and expressed as *P* values. *$P < 0.05$, **$P < 0.01$, ***$P < 0.001$. **j** Expression profiles of NUAK2 and YAP in HCC patient specimens and their adjacent normal specimens. $n = 233$ pairs, $P = 6.84E-15$ (paired Wilcoxon test). **k** Spearman correlation analysis of NUAK2 expression with YAP expression in HCC patient samples. $n = 247$, Rs = 0.34, $P = 3.2E-08$

and NUAK2 and the correlation of YAP and NUAK2 in hepatocellular carcinoma (HCC) through bioinformatics analyses. In support of our model, we found that NUAK2 expression levels are significantly up-regulated in a large cohort of HCC samples (Fig. 1j). Importantly, the level of *NUAK2* transcript was significantly and positively correlated with *YAP* mRNA abundance (Fig. 1k). Collectively, these findings lend

strong support to the YAP-driven NUAK2 misregulation in human HCC.

**NUAK2 is critical for YAP-driven liver growth and cancer.** To further demonstrate a role for NUAK2 downstream of YAP, we generated recombinant AAV2/8 viruses expressing two *Nuak2* or

*hYAP* targeting sgRNAs in addition to Cre recombinase (Fig. 2a). High titer administration of the Nuak2 sgRNA virus to TetO-YAP:Cas9 mice[40] resulted in a *Nuak2* knockout efficiency of approximately 70% from analysis of bulk liver DNA (Supplementary Fig. 2b). Considering there are still 30% of non-parenchymal cell types in the liver, which are not infected by the AAV2/8, our data actually underestimate the efficiency of Nuak2 deletion in hepatocytes, suggesting an almost complete knockout efficiency. We then administrated this AAV virus into adult mice bearing a Cre-dependent SpCas9-GFP transgene[40], and a Cre-dependent, Doxycycline (Dox)-inducible human YAP-S127A allele[1,41] (Fig. 2a) (to be referred as TetO-YAP:Cas9 hereafter). When a control AAV-Cre virus was injected into TetO-YAP:Cas9 mice, Dox induction led to massive and rapid hepatomegaly, as reported before[1,14]. Consistent with our previous data, YAP overexpressing livers exhibited higher NUAK2 expression at the mRNA and protein levels (Fig. 2d, e and Supplementary Fig. 2c). Strikingly, TetO-YAP:Cas9 mice infected with AAV-Cre-sgRNA-Nuak2 demonstrated a substantial and highly significant reduction in liver overgrowth and hepatocyte proliferation (Fig. 2b, c). To rule out nonspecific effects of the guide strand targeting, AAV-Cre-nontargeting sgRNAs was also included as control. Indeed, it displayed similar hepatomegaly phenotype as AAV-Cre. Most importantly, AAV-Cre-sgRNA-Nuak2 demonstrated a reduction in liver overgrowth (Supplementary Fig. 2a). The protein Myosin phosphatase target subunit (MYPT1) is currently the only validated substrate for NUAK2 kinase[42,43]. As predicted from our model, a significant increase in MYPT1 phosphorylation at serine 445, which is characterized as a NUAK2 phosphorylation site, was found in the livers of TetO-YAP:Cas9 mice infected with the control Cre virus. Phosphorylation of this substrate was abolished in the context of Nuak2 AAV-mediated knockout (Fig. 2d).

In addition to the short-term effects of *Nuak2* knockout in YAP-mediated overgrowth, we assessed the role of NUAK2 in a longer-term model of YAP-activation that leads to liver cancer. A mixed HCC/ICC pathology is generated by the expression of YAP-S127A for three months in a very small number of hepatocytes, achieved by the administration of a very low dose of AAV-Cre (Supplementary Fig. 2d). Strikingly, in this model, genetic ablation of *Nuak2* significantly suppressed the overall size, grade, and number of tumors formed (Fig. 2f, g). Taken together, our data strongly corroborate the essential role of NUAK2 in YAP-mediated hepatomegaly and tumorigenesis.

**Loss of *NUAK2* impairs growth in YAP-high liver cancer cells.** We next examined the requirement for NUAK2 in the proliferation of human cancer cells. Indeed, either *NUAK2* knockout by multiple sgRNAs or knockdown by shRNAs significantly suppressed proliferation of HuCCT-1 cells (Fig. 3a and Supplementary Fig. 3a, b). Importantly, growth inhibition in this context could be rescued by expression of a NUAK2 wild-type construct, but not by a kinase-dead mutant (Fig. 3b and Supplementary Fig. 3c), indicating the kinase dependent role of NUAK2 in controlling cell growth in vitro. Considering the strong functional relationship between YAP and NUAK2, we assessed whether liver cancer cell lines would show a selective response towards NUAK2 inhibition depending upon YAP/TAZ activity. We first selected several liver cancer cell lines with high- or low-YAP activity by determining YAP expression level (Fig. 3c), the activity of a TEAD/YAP transcriptional reporter[44] (Supplementary Fig. 3d), and the expression signature of YAP downstream targets (Supplementary Fig. 3e). From these data we cataloged HuCCT-1 and SNU475 cells as "YAP-high", whereas we considered HepG2 and SNU398 as having "YAP-low" activity. Furthermore, MYPT1, the

substrate of NUAK2, was highly phosphorylated at S445 in YAP-high cells, but not in cells with low YAP activity (Fig. 3c). As predicted, depletion of YAP/TAZ significantly inhibited the growth of YAP-high cells, while only negligible effects were observed in YAP-low cells (Fig. 3d and Supplementary Fig. 3f, g). Strikingly, YAP-high cells displayed a much higher sensitivity to NUAK2 knockdown than YAP-low cells (Fig. 3d and Supplementary Fig. 3h). In addition to using YAP-high and -low cells, we also utilized the cell lines with Hippo pathway mutations to demonstrate the role of NUAK2 on YAP-driven cell growth. As shown in Fig. 3e, the breast cancer cell line MDA-MB-231 and mesothelioma cell line NCI-H2052, cell lines with a known loss of NF2 expression, an upstream negative regulator of YAP, displayed higher phosphorylation of YAP on S127, compared to their counterpart wild-type NF2 cell lines. In line with our previous data, NUAK2 depletion significantly reduced the cell proliferation of the NF2 lost cell lines, while only negligible effects were observed in wild-type cells (Fig. 3f). Taken together, these data highlight NUAK2 inhibition as a potential selective dependency for YAP-driven malignancies.

**YAP-mediated NUAK2 induction auto-amplifies YAP activity.** Several previous reports have demonstrated that the actomyosin cytoskeleton regulates YAP activity[11,45–47]. As NUAK2 phosphorylates MYPT1 on S445, a regulatory subunit of the myosin light chain phosphatase (MLCP), it thereby inhibits phosphatase activity of MLCP to increase MLC phosphorylation with subsequent triggering of the assembly of actin fibers and creation of actomyosin tension[48–51]. Therefore, we hypothesized that YAP-mediated NUAK2 induction could form a double positive feedback loop to auto-amplify YAP activity through positively regulating actomyosin tension. Consistent with this hypothesis, knockdown of *NUAK2* reduced MYPT1 phosphorylation on S445 and promoted MLC dephosphorylation in vitro (Fig. 4a, b), which resulted in the disassembly of actin fibers (Fig. 4c). Depletion of NUAK2 led to the increased cytoplasmic accumulation of YAP (Fig. 4b–e) and the subsequent reduction of YAP transcriptional activity, as assayed by the mRNA expression of well-known YAP target genes, such as *CTGF*, *CYR61*, *AMOTL2*, and *ANKRD1* (Fig. 4f and Supplementary Fig. 4a).

Mechanistically, NUAK2 depletion also increased YAP cytoplasmic retention through moderately inducing YAP phosphorylation and 14-3-3 binding (Supplementary Fig. 4b, c), which is a well-known mechanism for YAP cytosolic retention[30]. However, whether NUAK2-mediated YAP translocation depends on the Hippo pathway still need to be further confirmed. More importantly, depletion of MYPT1 in the NUAK2-knockdown cells partially rescued not only cell growth but also YAP transcriptional activity (Supplementary Fig. 4d–f), indicating the involvement of MYPT1 in NUAK2 depletion-induced effects on cell growth inhibition and reduction of YAP activity.

To further examine the physiological YAP-NUAK2-MYPT1 axis in vivo, we also analyzed this signaling cascade in TetO-YAP:Cas9 mouse livers. YAP induction promoted NUAK2 expression with a consequential drastic increase in MLC phosphorylation, whereas Cas9-mediated viral *Nuak2* knockout attenuated this phenotype (Fig. 4g). Accordingly, NUAK2-depleted livers demonstrated reduced junctional actin fibers (Fig. 4h), which in turn resulted in less robust YAP nuclear localization (Fig. 4i) and significantly dampened transcriptional activity (Fig. 4j and Supplementary Fig. 4g).

If NUAK2 indeed participates in cell growth control via the regulation of YAP subcellular localization, we reasoned that forced translocation of YAP to the nucleus should rescue the phenotype of NUAK2 depletion. We tested this by stably

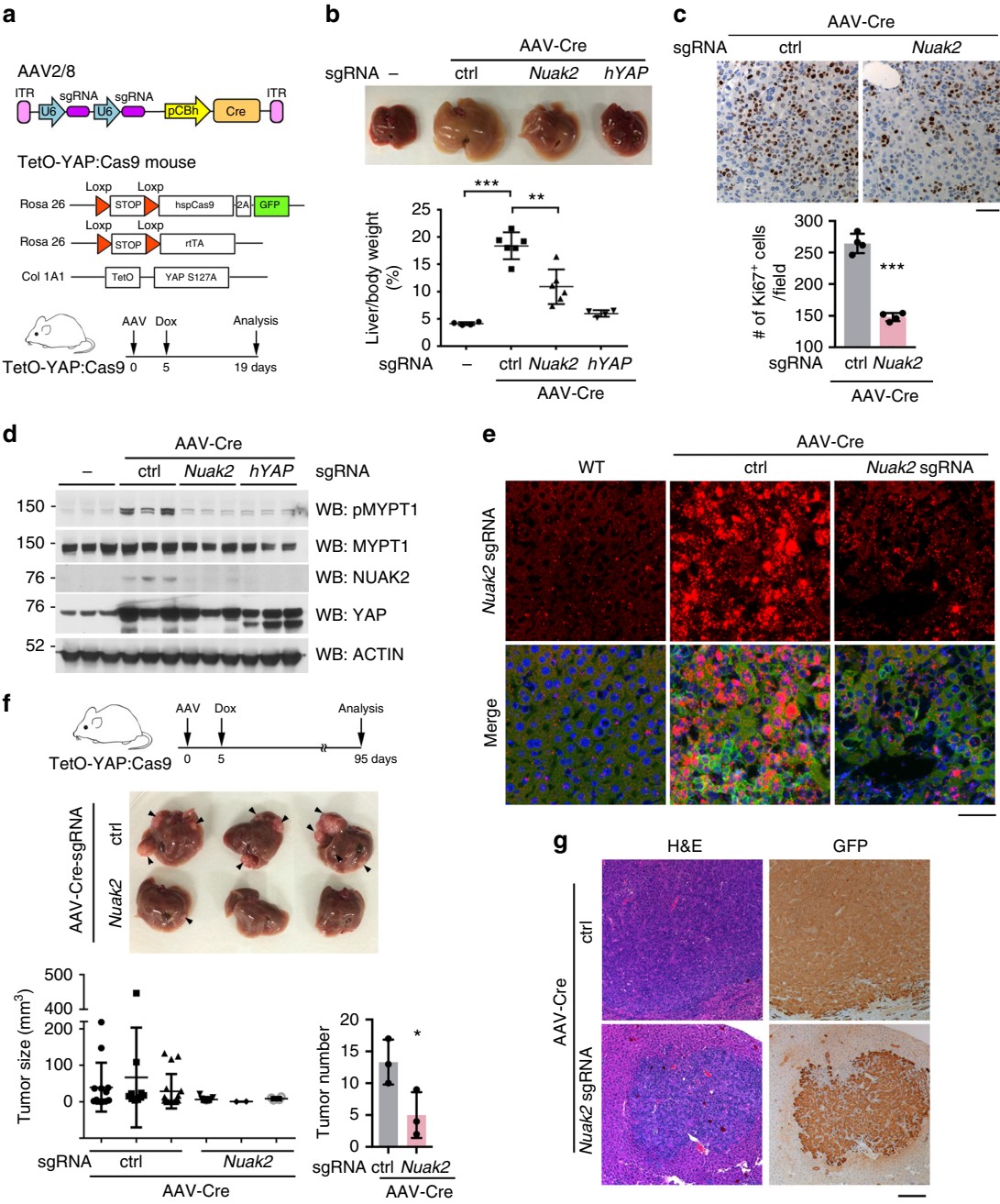

**Fig. 2** NUAK2 plays a critical role in YAP-driven hepatomegaly and tumorigenesis. **a** Top, schematic of AAVs utilized containing two sgRNAs targeting either *Nuak2* or transgenic YAP. Middle, transgenic mice bearing these three alleles were used for CRISPR/Cas9-mediated knockout of *Nuak2* in a YAP overexpression model. Bottom, experimental flow chart depicting protocol for YAP-mediated acute liver overgrowth. **b** Gross morphology of the livers of TetO-YAP:Cas9 transgenic mice infected with indicated AAV virus and placed on Dox for 2 weeks. Liver/body weight ratio of mice mentioned before was plotted. Data are presented as mean ± SD; $n = 4$ or 6. The two-tailed, Student's $t$ test was used to compare between two groups and expressed as $P$ values. $^{*}P < 0.05$, $^{**}P < 0.01$, $^{***}P < 0.001$. **c** Representative Ki67 staining of liver sections as in (**b**) and quantification of Ki67 positive cells were shown. Data are presented as mean ± SD; $n = 4$ mice per group, 5 or 6 high power field (HPF) per animal. The two-tailed, Student's $t$ test was used to compare between two groups and expressed as P values. $^{*}P < 0.05$, $^{**}P < 0.01$, $^{***}P < 0.001$. Bar, 25 μm. **d** Western blot analysis from livers of TetO-YAP:Cas9 mice as indicated. MYPT1 is a target of NUAK2 and it is used as a surrogate for its activity. Each lane represents a different mouse liver. **e** The designated liver tissue sections were analyzed by RNAscope for *Nuak2* (red) and co-stained with anti-GFP antibody (green) and DAPI (blue). Bar, 50 μm. **f** Experimental flow chart depicting protocol for long-term YAP-driven tumorigenesis. Gross morphology of livers in TetO-YAP:Cas9 transgenic mice infected with indicated AAVs. Arrowheads indicate visible tumor nodules. Bottom, size and number of tumors are indicated per histological analysis. Data are presented as mean ± SD; $n = 3$ animals. The two-tailed, Student's $t$ test was used to compare between two groups and expressed as $P$ values. $^{*}P < 0.05$, $^{**}P < 0.01$, $^{***}P < 0.001$. **g** Hematoxylin and eosin (H&E) and IHC analysis of liver tumor sections as in (**f**). Bar, 100 μm

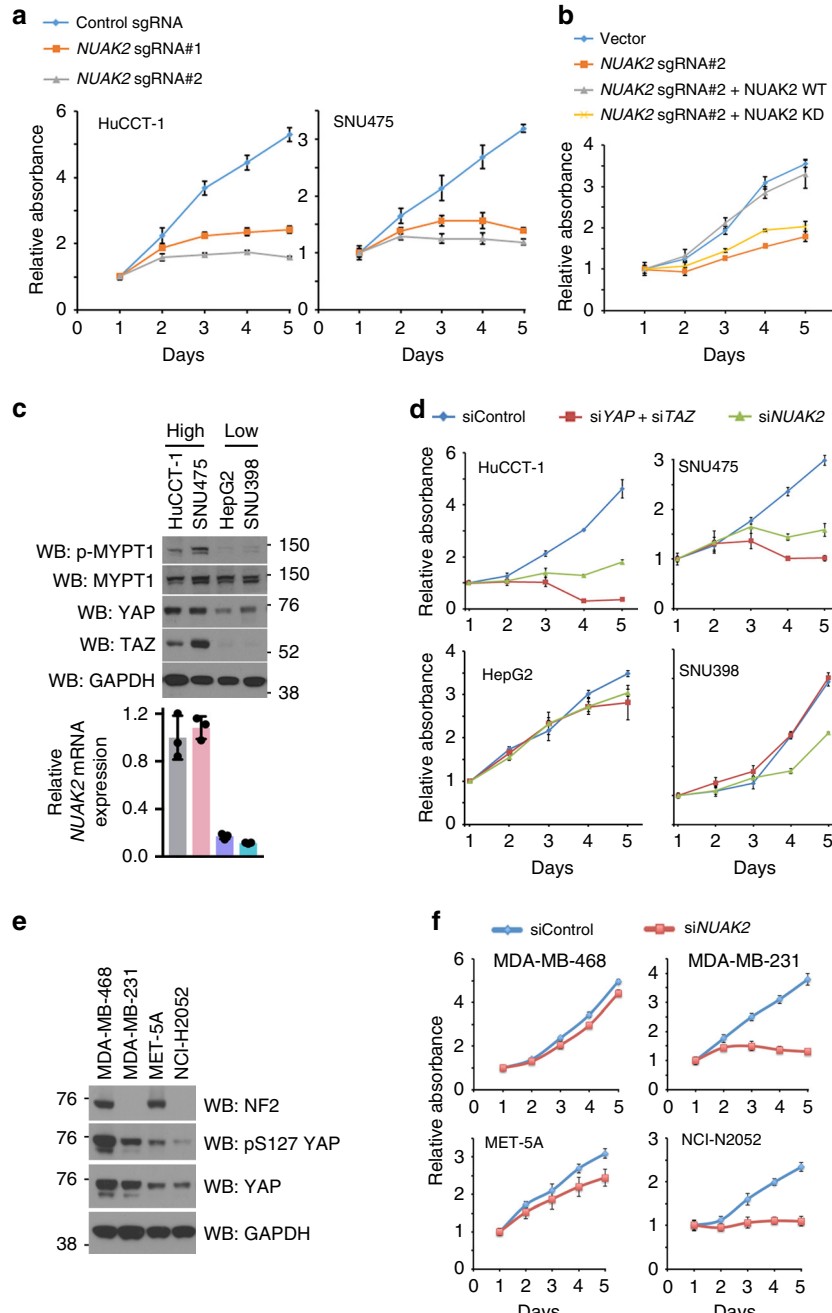

**Fig. 3** NUAK2-dependency in YAP-driven cancer lines. **a** Growth curves of HuCCT-1 and SNU475 cells transfected with either scrambled (ctrl) or NUAK2 sgRNAs using crystal violet assessment of cell growth. Data are presented as mean ± SD; $n = 6$. **b** Growth curves of HuCCT-1 cells transfected with scramble or NUAK2 sgRNA and/or vectors expressing cDNAs encoding wild-type or kinase-dead NUAK2 with a mutated PAM sequence, thus the construct DNA would not be targeted by Cas9. Data are presented as mean ± SD; $n = 5$. The expression of NUAK2 is shown in Supplementary Fig. 3c. **c** Western blot and qPCR analysis depicting levels of phospho-S445 MYPT1, MYPT1, YAP, TAZ, and NUAK2 in various human liver cancer cell lines. **d** Cell growth curves of liver cancer cell lines following transfection with siRNAs against YAP/TAZ, NUAK2 or control sequence. Data are presented as mean ± SD; $n = 4$. The knockdown efficiency of YAP, TAZ, and NUAK2 is shown in Supplementary Fig. 3f–h. **e** Western blot analysis depicting levels of NF2, YAP, and phospho-S127 YAP in indicated human cancer cell lines. **f** Cell growth curves of indicated cancer cell lines following transfection with siRNAs against NUAK2 or control sequence. Data are presented as mean ± SD; $n = 6$. The knockdown efficiency of NUAK2 is shown in Supplementary Fig. 3i

expressing a YAP-S127A construct carrying an ectopic nuclear localization signal (NLS), which should lead to constitutive nuclear translocation independent of cytoskeletal regulation. While the S127A mutation can also have an effect on nuclear localization, by itself it is not sufficient to confer full nuclear localization given that other Ser residues are also modified by LATS, and it is still subject to regulation by the cytoskeleton[11,30].

As predicted, the NLS-YAP-S127A mutant was largely accumulated in the nucleus, even in the absence of NUAK2, in contrast to the mostly cytoplasmatic localization of YAP-S127A (Supplementary Fig. 4h). Remarkably, NLS-YAP-S127A overexpression, but not YAP-S127A, partly restored growth inhibition caused by NUAK2 knockdown (Fig. 4k). Collectively, these data demonstrate that YAP-driven NUAK2 activity influences the

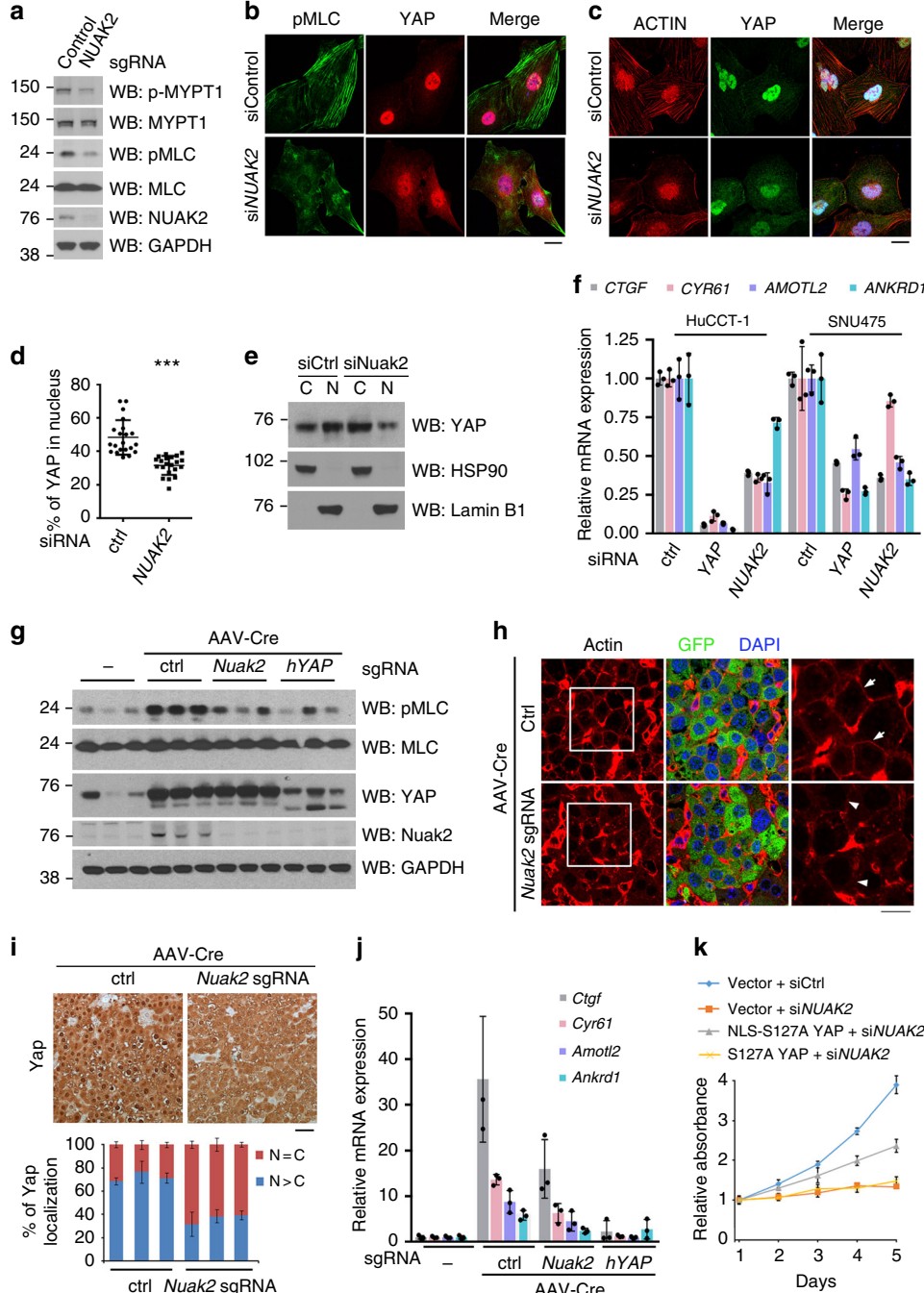

**Fig. 4** NUAK2 regulates YAP localization and activity via the actomyosin skeleton. **a** Western blot analysis of phospho-S445 MYPT1, MYPT1, NUAK2, phospho-MLC, and MLC in SNU475 cells transfected with indicated siRNAs. **b**, **c** Confocal analysis of SNU475 cells expressing indicated siRNAs and stained with DAPI and indicated antibodies. Bar, 20 μm. **d** The percentage of YAP localized in the nucleus was quantified in SNU475 shown in (**c**). Data are presented as mean ± SD; $n = 21$ HPF. The two-tailed, Student's $t$ test was used to compare between two groups and expressed as $P$ values. $^*P < 0.05$, $^{**}P < 0.01$, $^{***}P < 0.001$. **e** Cytosolic and nuclear fractions isolated from HuCCT-1 cells transfected with indicated siRNA were analyzed by Western blot. HSP90 serves as cytosolic marker, and Lamin B1 as nuclear marker. **f** qPCR analysis of YAP downstream genes in HuCCT-1 and SNU475 cells transfected with indicated siRNA. $n = 3$, mean ± SD. The knockdown efficiency of YAP and NUAK2 are shown in Supplementary Figure 4a. **g** TetO-YAP:Cas9 mice were infected with high-dose ($5 \times 10^{10}$ GC/mouse) AAV-Cre with the indicated sgRNA and administered Dox for 4 days. A piece of liver from three independent mice was analyzed by Western blot analysis. Each lane represents a different mouse liver. **h** The liver sections as in (**g**) were stained with anti-actin (red), anti-GFP (green) antibody, and DAPI. Arrows indicate the junctional actin; arrowheads point to the impaired actin bundles. Bar, 20 μm. **i** IHC analysis for YAP and relative quantification YAP localization. $n = 3$, mean ± SD. Bar, 20 μm **j** qPCR analysis of YAP downstream genes in TetO-YAP:Cas9 mice livers in g. $n = 3$ mice, mean ± SD. **k** Growth curves of HuCCT-1 cells stably expressing Dox-inducible YAP mutants transfected with scramble or NUAK2 siRNA. Data are presented as mean ± SD; $n = 8$

actomyosin cytoskeleton, which in turn forms a double positive feedback loop to maximize YAP activity.

**HTH-02-006 as a semispecific NUAK2 inhibitor**. Considering our observations suggesting that NUAK2 is critical for YAP-driven cell proliferation and its oncogenic phenotypes, the development of NUAK2 kinase inhibitors represents an attractive avenue for intervention in YAP-driven cancers. Here, we employed a novel semispecific inhibitor of NUAK2, HTH-02-006 (NUAK2 $IC_{50}$ = 126 nM), optimized from WZ4003[52] with improved selectivity to study the pharmacologic consequences of NUAK2 inhibition (Fig. 5a, b and Supplementary Fig. 5a). Indeed, HTH-02-006 treatment reduced levels of phosphorylated MYPT1 in HuCCT-1 cells indicating its activity on NUAK2 and/

or NUAK1, which can also phosphorylate MYPT1[43] (Fig. 5c). To examine the selectivity of HTH-02-006, we treated YAP-high and -low cell lines with increasing concentration of HTH-02-006. We find that YAP-high cell lines exhibit a notably higher sensitivity to the compound (Fig. 5d). To further demonstrate the specificity of HTH-02-006, we performed a protein sequence alignment of NUAK1 and NUAK2 and identified the A236 residue of NUAK2 that is conserved with the corresponding A195 residue of NUAK1, which is the binding site to WZ4003[52] (Supplementary Fig 5b). Based on the structural similarity of HTH-02-006 with its proto-type inhibitor WZ4003, we expected that the A236 site of NUAK2 would be crucial for successful binding of HTH-02-006. Indeed, bi-allelic knock-in $NUAK2^{A236T/A236T}$ cells generated by CRISPR gene-editing conferred partial resistance to HTH-02-006

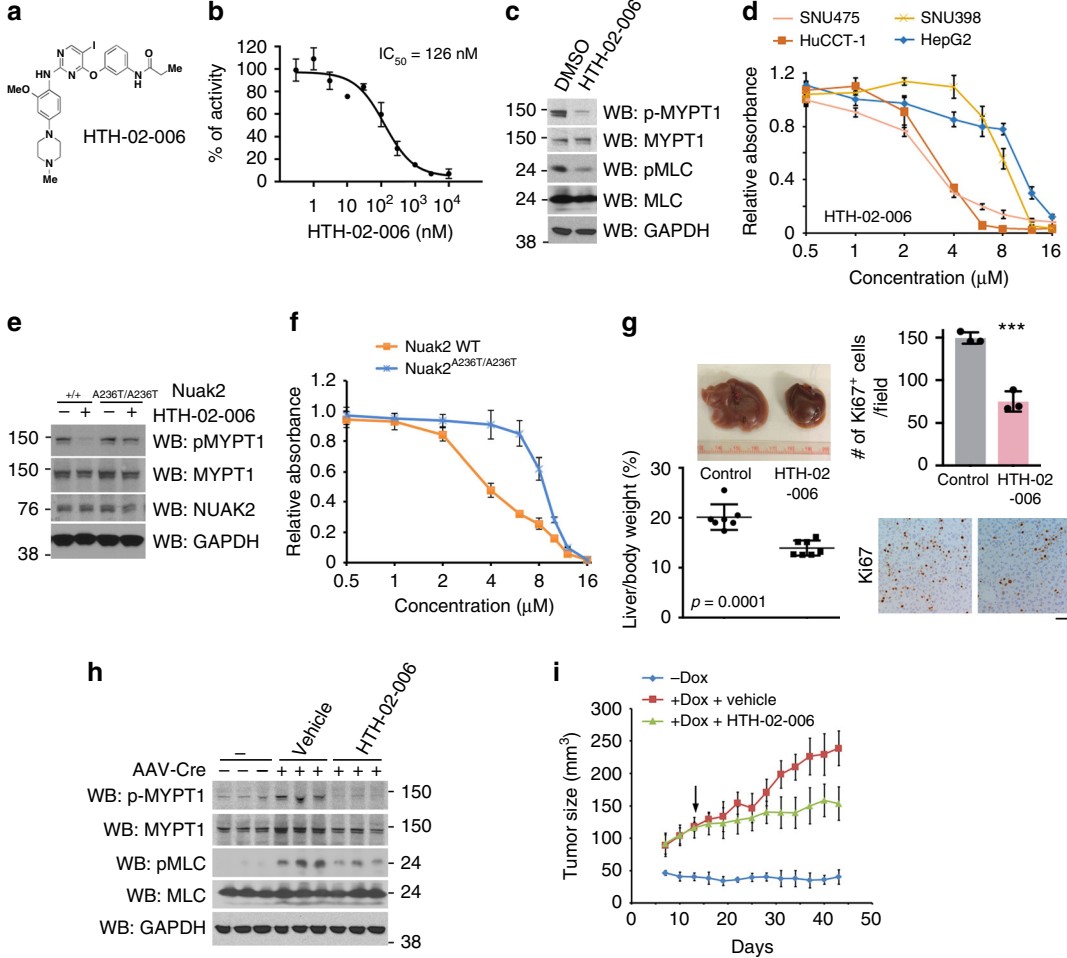

**Fig. 5** HTH-02-006 attenuates YAP-driven cell proliferation, hepatomegaly and tumorigenesis. **a** Chemical structures and IC50 of HTH-02-006. **b** In the presence of 100 μM [γ-32P]ATP, NUAK2 activity was analyzed using 200 μM Sakamototide with the indicated concentrations of HTH-02-006. The results are presented as the percentage of kinase activity relative to the DMSO-treated control. Results are mean ± S.D. $n$ = 2. **c** Western blot analysis of phospho-S445 MYPT1, MYPT1, phospho-MLC, and MLC in SNU475 cells treated with HTH-02-006. **d** Cell growth curves of indicated liver cancer cell lines. Cells were treated with increasing concentrations of HTH-02-006 for 120 h. Data are presented as mean ± SD; $n$ = 3. **e** Western blot analysis of indicated antibodies in wild-type and A236T mutant cells treated with/without HTH-02-006. **f** Growth curves of HuCCT-1 wild-type and A236T mutant cells treated with increasing concentrations of HTH-02-006 for 120 h. Data are presented as mean ± SD; $n$ = 3. **g** TetO-YAP S127A mice infected with AAV-Cre were fed Dox for 14 days. Data are presented as mean ± SD; $n$ = 7 animals per group. HTH-02-006 was administered by intraperitoneal injection twice a day during the 14 days period. Gross morphology of the liver in TetO-YAP S127A mice treated with HTH-02-006. Liver/body weight ratio of TetO-YAP S127A mice at the end of period is shown. Ki67 staining of liver sections and quantification of Ki67 positive cells are shown. Data are presented as mean ± SD; $n$ = 3, 5 HPF per mouse. The two-tailed, Student's $t$ test was used to compare between two groups and expressed as $P$ values. $^*P$ < 0.05, $^{**}P$ < 0.01, $^{***}P$ < 0.001. **h** Immunoblotting of liver lysates from TetO-YAP S127A mice treated for 14 days with vehicle or HTH-02-006. Three independent samples from each treatment were used. Each lane represents a different mouse liver. **i** Tumor growth of HuCCT-1 TetO-YAP S127A cells in a xenograft model. Nude mice were fed with or without Dox. HTH-02-006 (10 mg/kg) was administered by intraperitoneal injection twice a day. Data are presented as mean ± SD; $n$ = 4 animals per group

induced inhibition of MYPT1 phosphorylation, while MYPT1 phosphorylation was robustly downregulated in WT cells (Fig. 5e), suggesting A236T was a drug resistant mutant. In line with this notion, NUAK2 A236T cells were more resistant to HTH-02-006 induced cell growth inhibition, as compared to its counterpart WT cells (Fig. 5f), indicating that the effect of HTH-02-006 treatment is at least in part through inhibition of NUAK2 activity. Because HTH-02-006 does not only inhibit NUAK2 but also NUAK1 activity, we also tested whether NUAK1 is required for cell growth of HuCCT-1 cells. Compared to NUAK2, NUAK1 depletion exhibited negligible effects at similar knockdown efficiency, suggesting that the effect of HTH-02-006 is likely mediated through NUAK2, at least in this cellular context (Supplementary Fig. 5c).

**NUAK2 inhibition reduces YAP-driven liver growth and cancer.** To investigate the efficacy of the NUAK2 inhibitors in vivo, high-dose AAV-Cre receiving TetO-YAP S127A mice were injected intraperitoneally with HTH-02-006 (10 mg/kg) twice daily. At this dose, HTH-02-006 did not exert overt toxicity, as there was no evidence of body weight loss (Supplementary Fig. 5d). During the treatment period, mice were also administered Dox to activate YAP expression and to induce hepatocyte proliferation and consequent hepatomegaly. Compared with vehicle control, mice subjected to HTH-02-006 treatment exhibited potent suppression of YAP-induced liver overgrowth and a decreased number of proliferating hepatocytes (Fig. 5g). Furthermore, analysis of liver tissues also demonstrated dramatically decreased phosphorylation of MYPT1 at S445 (Fig. 5h).

Finally, we assessed the consequences of pharmacological NUAK2 in YAP-driven tumorigenesis. For this, we subcutaneously transplanted Dox-inducible YAP overexpressing HuCCT-1 cells into nude mice. Even though these cells exhibit nuclear YAP endogenously, exogenous expression of YAP permits faster growth of xenografts. When such tumors reached a volume of ~100 mm$^3$, mice were then treated with vehicle or HTH-02-006 (10 mg/kg) for 30 days. Continuous treatment of inhibitors did not affect body weight (Supplementary Fig. 5e), suggesting HTH-02-006 is well tolerated in nude mice. Compared to vehicle control, the growth rates of the tumors were significantly attenuated in HTH-02-006-treated mice (Fig. 5i and Supplementary Fig. 5e), indicating the antitumor activity of HTH-02-006. Together with our genetic data, our results strongly suggest that inhibition of NUAK2 kinase activity potently suppresses YAP-driven hepatomegaly and tumor growth.

## Discussion
The Hippo-YAP signaling pathway is thought to be a major contributor to tumorigenesis. Extensive data in animal models and clinical evidence support the idea that YAP/TAZ represent attractive therapeutic targets in cancer[18]. While some progress has been made in attempting to inhibit the YAP and TEAD interaction[25], as well as the stability of these molecules, because of the inherent limitations to such approaches there is still no specific and potent YAP/TAZ antagonist. Our data suggest an alternative way of targeting YAP, at least in the liver, which is by chemical inhibition of the NUAK2 kinase. Here, we provide genetic and chemical evidence that such interventions could represent a selective vulnerability of YAP-driven tumors.

NUAK2 is frequently amplified in a range of human cancers, forming part of the 1q32 amplicon common in melanoma, glioblastoma and other cancers[33,34,53]. Co-amplification of other oncogenes such as MDMX and the RAS-pathway effector ELK4 have, however, complicated the interpretation of the relevance of chromosomal gains containing NUAK2[54]. Further, heterozygous

knockout NUAK2 mice are sensitized to azoxymethane-induced colonic tumor formation[36]. Our data here are consistent with a clear growth-promoting function of NUAK2, at least in the hepatic context. Our work largely implicates myosin filaments and the actin cytoskeleton in the growth-promoting function of NUAK2. However, it is important to note that very little is known about the other targets of this kinase, and it is likely that other substrates might also play an oncogenic role downstream of NUAK2. For instance, it has been reported that mTOR activation and CDK2 induction are associated with NUAK2 expression[34,55,56] though the underlying mechanisms remain unclear. Clearly, elucidation of the full set of NUAK2 substrates will clarify this issue. Additionally, several inputs including nutrient and oxidative stress have been shown to act upstream of NUAK2[57,58]. To what extent these can impinge upon YAP activity and growth remain to be determined.

Targeting the Hippo pathway has become an intriguing avenue for cancer therapeutics. One major limitation to move this forward has been the lack for traditionally druggable molecules in the pathway. While inhibitors for Mst1/2 kinases have been reported[59], these inhibitors are likely to be useful in a pro-regenerative context, but not in a cancer setting, given the known growth suppressive function of Mst1/2. Interfering with the interactions of YAP with TEAD or other regulatory factors is another approach, although the lack of enzymatic pockets in these proteins will require the development of antagonists of protein–protein interfaces, an strategy that has been historically challenging[20]. Our data here provide multiple lines of evidence that inhibitors of the kinase activity of NUAK2 represent a novel approach to modulate YAP function in vivo. Moreover, our data suggest that NUAK2 inhibition might represent a selective vulnerability of cancer cells with high YAP activity. It is unclear at this point, whether this finding is limited to liver cancer, or whether it could be applicable to other malignancies. Additionally, it will be important to identify particular cancer genotypes that would correlate with sensitivity to NUAK2 depletion. In any case, the novel compound described in this report, could be the basis for the development of more specific and potent inhibitors.

In addition to its aberrant role in tumorigenesis, the Hippo-YAP signaling cascade is physiologically involved in the control of many basic biological functions. Therefore, the dynamics of YAP/TAZ activity have to be tightly controlled to ensure the proper physiological functions. Several negative feedback regulatory mechanisms control YAP activity to prevent over-activation[60–62]. In this study, we elucidate a mechanism by which YAP-mediated NUAK2 induction amplified YAP activity via the cytoskeleton. Together with other positive regulatory mechanisms[63], YAP activity is able to be amplified rapidly and robustly to ensure a timely adjustment during acute conditions, like tissue injury, thereby maintaining tissue homeostasis. However, tumor cells can also hijack these pathways to drive uncontrolled cell growth. It might also explain the dominant YAP-activation phenotypes in many cancer types without genetic aberrations of the pathway.

## Methods
**Cell culture and establishment of stable cell lines.** All cell lines were obtained from American Type Cell Collection (ATCC, USA). HuCCT-1, SNU475, SNU398, MDA-MB-231, MDA-MB-468, and NCI-H2052 cells were cultured in RPMI containing 10% FBS, 1× Hepes, 1× L-glutamine. HepG2 and 293T cells were cultured in Dulbecco modified Eagle's medium (DMEM) containing 10% FBS. H69 were maintained in DMEM: DMEM/F12 containing 10%FBS and supplemented with Adenine, Insulin, Epinephrine, T3-T, EGF and hydrocortisone as previously reported[64]. MET-5A were maintained in Medium 199 containing 10%FBS and supplemented with HEPES, insulin, EGF, trace elements B and hydrocortisone. HuCCT-1 and H69 TetO-YAP stable cells were generated by lentiviral transduction of a plasmid expressing M2-rtTA construct and a pNL-TRE-YAPS127A plasmid. HuCCT-1 NLS-YAP S127A and YAP-S127A were generated by lentiviral

transduction. *NUAK2* shRNAs were obtained from GE Dharmacon (RHS4533-EG81788). siRNAs were obtained from Ambion (*YAP* s20367, *TAZ* s24787, *NUAK2* s37779, and *MYPT1* s9235) and Sigma (*NUAK1* SASI_Hs01_00188391). All plasmid transfections were performed with Lipofectamine 3000 and siRNA transfections were performed with Lipofectamine RNAiMAX according to the manufacturer's recommendations.

**Plasmids.** NUAK2 cDNA was cloned to pRK5-Flag and NUAK2 T252E (kinase-dead) and G9A mutants were generated by site-directed mutagenesis.

**Immunofluorescence.** Cells were fixed with 4% paraformaldehyde/phosphate-buffered saline (PBS) for 10 min at room temperature, followed by three washes in PBS. Cells were permeabilized in 0.01% Triton/PBS for 1 min, followed by three washes in 0.01% Tween/PBS and then blocked with PBS supplemented with 10% donkey serum for 1 h. Cells were incubated with primary antibody (YAP (Cat#14071, 1:200, Cell signaling), pMLC (Cat#3671, 1:100, Cell signaling), actin (Cat#A5316, 1:200, Sigma) and Flag (Cat#F3165, 1:400, Sigma)) in blocking buffer for overnight. After three washes in 0.1% Tween PBS, cells were incubated in secondary antibody and 1 μg/ml DAPI for 1 h at room temperature. Cells were then washed, mounted and examined with a confocal microscope (Zeiss LSM 700 Laser Scanning Confocal), equipped with a 25× or 40× oil objective lens or epimicro-scope (Zeiss Axio Observer Z1).

**Immunohistochemistry.** Tissue was fixed in 4% paraformaldehyde (PFA) for 48 h and then embedded in paraffin for sectioning. Tissue sections measuring 5 μm were deparaffinized and rehydrated followed by antigen retrieval using low-pH Antigen Unmasking Solution (Vector Labs, Burlingame, CA). Quenching of endogenous peroxidase (0.3% H$_2$O$_2$) and protein block (10% Donkey serum) was performed prior to overnight primary antibody incubation at 4 °C (GFP (Cat#ab6673, 1:1000, Abcam), YAP (Cat#14071, 1:400, Cell Signaling), Ki67 (Cat#GTX166667, 1:200, GeneTex), Sox9 (Cat#AB5535, 1:200, Millipore), actin (Cat#A5316, 1:100, Sigma)). Sections were washed and Vectastain Elite ABC kit and secondary antibody (Vector Labs) were used to detect primary antibody binding. Slides were developed using the Vectastain Elite kit used as directed by the manufacturer. Counterstaining was done with haematoxylin and samples were washed, dehydrated, and mounted with Vectamount (Vector Labs #H-5000).

**Immunoblotting.** Cells and tissue were collected, and processed for western blotting by solubilizing extracts in radioimmunoprecipitation assay (RIPA) buffer with protease inhibitor cocktail (Roche #04693159001) and phosSTOP (Roche, # 04906837001)). Protein lysates were resolved by polyacrylamide gel electrophoresis (PAGE) under reducing conditions (4–12% sodium dodecyl sulfate–PAGE Bis–Tris gels; MOPS buffer system; Invitrogen; NuPAGE-MOPS system). The gels were blotted onto nitrocellulose or PVDF membranes and blocked in either 5% milk or 1% bovine serum albumin and followed by primary antibody incubation for overnight (NUAK2 (Cat#sc-374348, 1:100, Santa Cruz), YAP (Cat#14071, 1:1000, Cell Signaling), GAPDH (Cat#3683, 1:10000, Cell Signaling), actin (Cat#A5316, 1:1000, Sigma), pMLC (Cat#3671, 1:1000, Cell Signaling), MLC (Cat#M4401, 1:1000, Sigma), TAZ (Cat#4883, 1:1000, Cell Signaling), NF2 (Cat#12888, 1:1000, Cell Signaling), MYPT1 and p-MYPT1 (S445) antibodies were a kind gift from Dr. Dario Alessi[43]. After washing with TBST, membranes were incubated with horseradish peroxidase-conjugated secondary antibodies for 1 h. Blots were washed with TBST and developed with the enhanced chemiluminescence system. Uncropped Western blot images of data shown in Figures can be found in Supplementary Figure 6.

**Immunoprecipitation.** HuCCT-1 cells were transfected with the indicated expression vectors and siRNA. After 72 h transfection, cells were lysed in RIPA buffer with protease (Roche) and phosphatase (Roche) inhibitor. 1 mg of total lysates were precleared for 30 min at 4 °C, and then immunoprecipitated with anti-HA beads (Sigma) overnight at 4 °C. The immunoprecipitates were washed with RIPA buffer three times.

**Cell fractionation.** Cells were transfected with indicated siRNA and incubated for 72 h, and then cells were lysed and fractionated into nuclear and cytosolic fraction using NUCLEI EZ PREP NUCLEI ISOLATION KIT (Sigma) following the manufacturer's recommendation.

**Cell proliferation assay.** Cells were transfected with indicated siRNA and incubated for 24 h, and then reseeded into 96-well plates at a density of 2000 cells per well. At indicated time points, cells were fixed (4% PFA) and stained with crystal violet. The dye was then extracted by adding 10% acetic acid to each well, and the absorbance was measured at 600 nm.

In order to test the inhibitors, cells were seeded at a density of 2000 cells per well in 96-well plates and on the next day treated with HTH-02-006 in the specific concentrations. After 5 days incubation, cells were fixed and stained with crystal violet. The dye was then extracted by adding 10% acetic acid to each well, with absorbance measured at 600 nm.

**Mouse models.** Animal work was approved by the institutional committee at Boston Children's Hospital. Animals were housed in specific pathogen-free facilities at the hospital. Tetracycline-inducible YAP S127A expression mice were previously described[1]. Cre-dependent Cas9 mice were obtained from Jackson Laboratories (Bar Harbour, MA) B6;129-Gt(ROSA)26Sor < tm1(CAG-cas9*,-EGFP)Fezh > /J. AAV2/8-sgRNA-Cre viruses were generated by Gene Transfer Vector Core in Harvard Medical School. The specific AAV viruses were injected retro-orbitally. For TetO-YAP S127A overexpression, 3 days after AAV-Cre delivery, mice were administered 2 mg/ml doxycycline ad libitum in their water.

To administer inhibitors to mice, HTH-02-006 was dissolved in 5% Dextrose and 10% DMSO. HTH-02-006 was administered by intraperitoneal route at a dose of 10 mg/kg for HTH-02-006 twice a day.

For tumor growth in the xenograft model, 8-week-old Nude mice (Jax stock #00785) were injected s.c. with 5 × 10$^6$ HuCCT-1 TetO-YAP S127A cells (n = 4 for each group) mixed with 100 μl RPMI medium and 100 μl Matrigel (BD). Until tumor size of Dox administrating mice around 100 mm$^3$, mice were randomly divided into two groups. Mice were then given vehicle or HTH-02-006 for 30 days.

**Luciferase assays.** Cells were cotransfected with a pGL3-based reporter construct and pCMV-Renilla. Luciferase activity in cell lysates was assayed by the Dual Luciferase Reporter Assay System (Promega). The relative promoter activity was present as the fold-change in firefly luciferase activity after normalization to the renilla luciferase activity.

**qPCR analysis.** Total RNA was extracted from cell pellets using Trizol (Life Technologies) according to the manufacturer's instructions. cDNA was synthetized using iScript Reagents (Biorad). Quantitative polymerase chain reaction (qPCR) was performed with the One Step plus Sequence Detection System (Applied Biosystems) using Fast SYBR green master mix reagent (Applied Biosystems) or TaqMan-based Real-Time PCR gene expression assays (*Applied Biosystems*).

Gene expression levels were normalized to a housekeeping gene. qPCR primers sequences are listed in Supplementary Table 2.

Taqman assays for mouse and human YAP1 (Mm00494240_m1, Hs00902712_g1), TAZ (Hs00210007_m1), CTGF (Mm01192933_g1, Hs01026927_g1), Cyr61 (Mm00487499_g1, Hs00964221_g1), AMOTL2 (Mm00502287_m1, Hs01048101_m1), and ANKRD1 (Mm00496512_m1, Hs00173317_m1) were multiplexed with 18S control assays (Applied Biosystems).

**RNA-seq.** AAV-Cre was given to TetO-YAP mice retro-orbitally. After 3 days, mice were administered Dox ad libitum in their cage water. Five days after Dox administration, hepatocytes were isolated and RNA was purified. RNA-seq libraries were generated using TruSeq RNA Sample Prep Kit v2 (Illumina) according to manufacturer's recommendations. The high-throughput sequencing was carried out on HiSeq 2000 (Illumina) at the Tufts Genomics Core.

**ChIP-seq.** Four days after Dox administration, hepatocytes were isolated and purified from TetO-YAP mice. Hepatocytes were cross-linked in 1% formaldehyde for 10 min at room temperature after which the reaction was stopped by addition of 0.125 M glycine. Cells were then lysed in ChIP buffer (100 mM Tris at pH 8.6, 0.3% SDS, 1.7% Triton X-100, and 5 mM EDTA) and the chromatin was sheared with a Diagenode Bioruptor sonicator UCD-200 to obtain fragments of average size of 200-500 bp. Suitable amounts of chromatin were incubated with antibodies for YAP (Cell Signaling #14074), TEAD4 (Abcam ab58310), or H3K27ac antibody (Abcam ab4729) overnight. Complexes were recovered on Protein-A/G agarose beads (Pierce) and, after multiple washes, DNA was reverse cross-linked and purified using the QIAquick PCR purification kit (QIAGEN). Libraries for ChIP-sequencing were generated by using the NEB Next Ultra DNA Library Prep Kit for Illumina (NEB) and barcoded using NEB Next Multiplex Oligos for Illumina (Index Primers Set 1) (NEB) according to the manufacturer's instructions.

**ChIP-Seq data analysis.** Fastq files were generated from ChIP-seq of mouse liver or retrieved from GEO (GSE68296, GSE62275, GSE66083, and GSE61852). ChIP-Seq data analysis was done using the docker4seq package (https://github.com/kendomaniac/docker4seq), part of reproducible-bioinformatics project (www.reproducible-bioinformatics.org). Illumina adapters were removed using skewer[65]. Trimmed reads were mapped on the corresponding reference genome (hg38, mm10) using BWA[66]. Transcription factors peaks were detected using MACS[67] and histone marks using SICER[68]. The association of peaks to genes was done using ChIPPeakAnno[69], Bioconductor package. All analyses were done using the default parameters set in docker4seq chipseqCounts function. BigWig generation and normalization was done using bedtool v2.17.0[70]. Enhancers were defined as regions positive for overlapping H3K27ac and H3K4me1 signals. Super enhancers were defined as clusters of enhancers within 12.5 kb one from another as default parameter of the ROSE software, as previously described (Hnisz et al., 2013). Input constituents used were H3K27Ac$^+$/H3K4Me1$^+$ peaks (putative enhancers) with a ranking BAM file containing H3K27ac reads, and a control BAM file containing IgG reads.

**General information for chemical synthesis**. Unless otherwise noted, reagents and solvents were obtained from commercial suppliers and were used without further purification. [1]H NMR spectra were recorded on Bruker A500 (500 MHz), and chemical shifts are reported in parts per million (ppm, δ) downfield from tetramethylsilane. Coupling constants ($J$) are reported in Hz. Spin multiplicities are described as s (singlet), br (broad singlet), d (doublet), t (triplet), q (quartet), and m (multiplet). Mass spectra were obtained on a Waters Micromass ZQ instrument. Preparative HPLC was performed on a Waters Sunfire C18 column (19 × 50 mm, 5 μM) using a gradient of 15–95% methanol in water containing 0.05% tri-fluoroacetic acid over 22 min (28 min run time) at a flow rate of 20 mL/min. Purities of assayed compounds were in all cases greater than 95%, as determined by reverse-phase HPLC analysis. Details of chemical synthesis and the relevant NMR spectra are included in Supplementary Method 1.

**$IC_{50}$ determination**. HEK-293 cells were transfected with pEBG2T mammalian constructs expressing N-terminal GST- tagged NUAK2. Active GST–NUAK2 was purified by glutathione–sepharose[52]. For peptide kinase assays, each reaction was performed in duplicate in 96-well plates. Totally, 100 ng of NUAK2 was incubated in 50 mM Tris/HCl (pH 7.5), 0.1 mM EGTA, 10 mM magnesium acetate, 200 μM Sakamototide, 0.1 mM [γ-32 P]ATP (450–500 c.p.m./pmol) and the indicated concentrations of HTH-02-006 dissolved in DMSO for 30 min at 30 °C. Reactions were terminated by adding 25 mM (final) EDTA to chelate the magnesium. Then, 40 μl of the reaction mix was spotted on to P81 paper and immersed in 50 mM orthophosphoric acid. Samples were washed three times in 50 mM orthophosphoric acid followed by a single acetone rinse and air drying. The incorporation of [γ-32 P]ATP into Sakamototide was quantified by Cerenkov counting. The values were expressed as a percentage of the DMSO control.

**CRISPR/Cas9 gene-editing approach**. Construction of lenti-CRISPR/Cas9 vectors targeting NUAK2 in human cell lines was performed following the protocol associated with the backbone vector (Addgene, #52961). The software (http://crispr.mit.edu/) predicted the following sequences with priority given to sequences that matched the early coding exons of targeted genes. The guide sequence used are 5′-TGGAGTCGCTGGTTTTCGCG-3′ or 5′-CTGTGCTTTACTGCGCGCTC-3′ for human NUAK2.

For *NUAK2* A236T knock-in cell line generation, the guide sequence used is 5′-GTGCAGGATTGAGTCAAACAC-3′ and the ssODN sequence used is 5′-TGG GTTTCCCCAGGGTCTGGAGAAGGGCGGGCCAAGAGCTGAAGAAAAC CCAGGCCTTGACATGACCTGTGACCGTGTTTGACTCAATCCTGCAGATTA CCGACTTCGGTCTCTCCAACCTCTACCATCAAGGCAAGTTCCTGCAGACA TTCTGTGGGAGCCCCCTCTATGCCTCGCCAGAGATTGTCAATGGGAAG CC-3′.

AAV-sgRNA-Cre plasmid is modified from AAV:ITR-U6-sgRNA(backbone)-pCBh-Cre-WPRE-hGHpA-ITR plasmid which was a gift from Feng Zhang (Addgene #60229)[40]. The guide sequence used are listed below
Non-targeting, 5′-ATGTTGCAGTTCGGCTCGAT-3′ and 5′-ACGTGTAAGGCGAACGCCTT-3′, mouse Nuak2, 5′-ATGGTGCGGGGACCGCGAGG-3′ and 5′-CTACGAGTTCCTGGAGACGC-3′, human YAP, 5′-CGACTCCTTCTTCAAGCCGC-3′ and 5′-GCCGGTTGCCCGGGTCCGGA-3′.

**NUAK2 and YAP gene expression analysis**. We downloaded gene expression data of previous study[71] from GEO database (GES14520). Then, *NUAK2* and *YAP* expression of all liver tumor samples were extracted to perform Spearman correlation analysis. Correlation coefficient and p-value were reported. To compare the gene expression level in tumor versus normal tissues, we used paired Wilcoxon test to compare the matched tumor-normal gene expression data of *YAP* and *NUAK2*.

**Statistical analysis**. The two-tailed, Student's $t$ test was used to compare between two groups and expressed as P values. $*P < 0.05$, $**P < 0.01$, $***P < 0.001$.

## Data availability

The Gene Expression Omnibus accession number is [GSE107860]. Additional data supporting the findings of this study are available from the authors on request.

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

## Acknowledgments

We are grateful to all the members of the Camargo lab for insightful discussion and support. We thank Dario R. Alessi and Sourav Banerjee for determination of the IC50 of HTH-02-006 for inhibiting NUAK2. We are indebted to Nathanael S. Gray and Tinghu Zhang for chemistry advice. This study was supported by grants from the National Institutes of Health (AR064036 and DK099559 to F.D.C.). F.D.C. was a Pew Scholar in the Biomedical Sciences. W.-C.Y. was supported in part by the Postdoctoral Research Abroad Program fellowship, Taiwan National Science Council (NSC), and National Cancer Center (NCC) CHILDREN'S CANCER PROJECT postdoc fellowship. M.T.D. was supported by Swiss National Science Foundation fellowship (P2BSP3_161941).

## Author contributions

W.-C.Y. and F.D.C. designed experiments and wrote the manuscript. W.-C.Y., B.P.-M., G.G.G., and M.T.D. performed the experiments and data analysis. H.-T.H., and M.H. designed and synthesized the inhibitors. H.L., Y.W. and R.A.C. performed all bioinformatic analyses. All authors critically discussed the results and the manuscript. F.D.C. supervised the project and gave final approval.

## Additional information

**Competing interests:** The authors declare no competing interests.

