## [Peer Review File · Nature Communications]

Reviewers' Comments:

Reviewer #1:

Remarks to the Author:

Here the authors identified NUAK2, a member of the AMPKR family of kinases, as a YAP target gene in murine hepatocytes, engineered to inducibly express activated YAP S127A, based on an increased level of TEAD4 transcription factor binding to a regulatory element downstream of the Nuak2 gene. Analysis of a series of human cancer lines with activated YAP also showed YAP/TEAD binding to the NUAK2 gene, and knockdown of YAP reduced NUAK2 expression. Consistently, the levels of NUAK2 and YAP RNAs were positively correlated in human HCC samples. AAV-mediated expression of a Nuak2 sgRNA in the liver of Cre-dependent Cas9 mice bearing a Cre-dependent, Dox-inducible human YAP-S127A allele greatly reduced the hepatomegaly and hepatocyte proliferation that is normally induced in these mice by Dox treatment, concomitant with reduced Nuak2 expression and phosphorylation of MYPT1, a known NUAK2 substrate that acts a regulatory subunit of PP1 to target phospho-MLC. NUAK2 phosphorylation of MYPT1 reduces MYPT1/PP1 phospho-MLC phosphatase activity. When AAV-Cre was administered at a low dose to induce YAP-S127A in a small number of hepatocytes, Nuak2 sgRNA-mediated depletion of Nuak2 reduced the number and size of liver tumors. NUAK2 depletion in HCC cells with high YAP levels, such as human HuCCT-1 cells, reduced proliferation of these cells, the expression of YAP/TEAD target genes and MYPT1 phosphorylation. These phenotypes were reversed by expression of WT but not kinase-dead NUAK2. Loss of NUAK2 resulted in decreased phosphorylation of MYPT1, which in turn increased dephosphorylation of phospho-MLC by MYPT1/PP1. The resultant decrease in MLC phosphorylation led to disassembly of actin fibers, which in turn reduced YAP expression that is known to be stimulated via assembly of the actin cytoskeleton. The same effects of Nuak2 depletion were observed in mouse livers, where there were reduced junctional actin fibers and nuclear YAP. Expression of nuclearly targeted NLS-YAP-S127A partly reversed the decreased proliferation of HuCCT-1 cells observed with NUAK2 knockdown. Finally, they showed that HTH-02-006, a semi-specific inhibitor of NUAK2 kinase, selectively inhibited proliferation of cancer cell lines with high YAP, and that HTH-02-006 inhibition of NUAK2 suppressed YAP-driven hepatomegaly and tumorigenesis in the Dox-inducible YAP-S127A mouse model.

This is well conducted study convincingly demonstrating that the NUAK2 protein kinase, a member of the AMPKR family of kinases, is a target gene for the Hippo growth regulatory pathway, whose role is to inhibit proliferation and growth by phosphorylation of the YAP transcriptional co-activator protein to retain it in the cytoplasm and prevent it from activating expression of proliferation-related genes. Their data show that NUAK2 is a direct target for YAP/TEAD, and that disruption of YAP in tumor cells with high YAP-expression decreases proliferation in culture, and YAP-S127A-driven hepatocellular carcinoma in mice. The loss of NUAK2 kinase activity leads to increased MYPT1/PP1 phosphatase activity, which reduces MLC phosphorylation and decreases actin polymerization, leading in turn to loss of a feedback loop that would normally trigger further YAP expression.

One major question raised by these studies is whether MYPT1 is the most important or only substrate for NUAK2 relevant to the cell proliferation and tumorigenesis phenotypes, and there is no discussion of this issue. The alteration in the actin cytoskeleton when NUAK2 expression is reduced is consistent with NUAK2-mediated MYPT1 phosphorylation playing a role, but there certainly could be other NUAK2 substrates that are also important for tumorigenesis.

Points: 1. Figure 2d: While the increase in MYPT1 phosphorylation was strong, the increase in Nuak2 protein was modest. Presumably each lane represents a different mouse liver, and this should be indicated in the figure legend.

2. Figure 2: There are many phosphorylation sites mapped in MYPT1, and the authors need to indicate in the Results that have monitored phosphorylation of S445, which is a characterized NUAK2 site.

3. Figure 3: The authors should explain why Cas9 does not target the NUA2 sgRNAs to act on the co-transfected WT and KD NUA2 construct DNAs.

4. Figure 4a/b: There was a partial reduction of nuclear YAP upon NUA2 depletion (was this quantified and confirmed by cell fractionation?), but the mechanism is not immediately obvious. The authors imply that this was due to the reorganization of the actin cytoskeleton, but was there also decreased phosphorylation of S127 and 14-3-3 binding (this should be checked) or is another mechanisms involved.

5. Figure 5: Does HTH-02-006 treatment of hepatoma lines reduce YAP protein expression, e.g. by blocking feedback, or decrease NUA2 levels (some inhibitors lead to decreased kinase protein stability)? As the authors suggest, the inhibitory effect of HTH-02-006 treatment on xenograft tumor growth could primarily be due to inhibition of NUA2 activity, but given the possible contribution of off-target effects, what is really needed is re-expression of an HTH-02-006-resistant mutant form of NUA2 in these cells to demonstrate that tumor growth is now unaffected by HTH-02-006 treatment.

Reviewer #2:

Remarks to the Author:

Yuan et al studied the mechanism of YAP in hepatomegaly and tumorigenesis. The authors discovered that NUA2 as a direct downstream target gene of YAP. Importantly, knockdown of NUA2 significantly blocked the hepatomegaly and liver tumors that were induced by active YAP expression. The authors also showed that NUA2 formed a positive feedback loop to activate YAP. Furthermore, they identified NUA2 inhibitors, though not very specific, and showed that pharmacological inhibition of NUA2 could suppress cell growth of YAP high cells as well as liver tumors induced by active YAP expression.

This study identifies a critical downstream target gene, NUA2, in mediating the physiological function of YAP. Moreover, their data suggests that NUA2 could be therapeutic target for YAP-driven cancers. Therefore, the study is interesting and significant for both basic research to understand the mechanism of YAP in cell growth regulation and potential targets for cancer drug development.

One major remaining issue is whether the NUA2 and its inhibitor HTH-02-006 selectively inhibit growth of YAP dependent cells. Though the authors have tried to address this issue in Fig.5d, but the data is rather weak. The authors test NUA2 knockdown and HTH-02-006 against cancer cells that the Hippo pathway are mutated, such as mutation in NF2 or LATS, to show the specific role of NUA2 in YAP high cell. Alternatively, the authors can use wild type and LATS knockout MEF to demonstrate whether NUA2 knockdown or HTH-0-006 is more sensitive to YAP high cells.

Fig.3b, 3d. The NUA2 knockdown efficiency, such as Western of NUA2 protein, should be included. NUA2 is a YAP target gene, but its mRNA is very low in the HuCCT-1 cells. Do the authors have explanation for this?

Fig.4a. Again, Western for NUA2 should be presented to show the knockdown efficiency.

Reviewer #3:

Remarks to the Author:

The authors identified an AMPK-related kinase, NUA2, as a potential transcriptional target of the Hippo-YAP signaling pathway. The authors note that this is an important endeavor because the

Hippo pathway is difficult to access therapeutically and that there are not that many bona fide direct targets that are amenable to chemical inhibition. They convincingly show that YAP activates NUA2 through TEAD-DNA-binding. NUA2 expression is upregulated in liver cancer cells and is correlated with YAP expression. Mechanistically, NUA2 acts on Myosin phosphatase target subunit (MYPT1), increasing MLC phosphorylation and triggering the assembly of actin fibers. They propose that this forms a positive feedback loop to auto-amplify YAP activity through increased actomyosin tension. Targeting NUA2 by HTH-02-006 attenuates cancer cell line proliferation and progression.

The work is clear and solid, the models are varied and robust, and ultimately I am convinced that this is a good target of Yap and worthy of therapeutic intervention. There are some things that can be clarified or supplemented so that the work can be better interpreted and repeated:

1. The in vivo AAV-Cas9 experiment to delete NUA2 is very elegant (fig. 2), however, there are many technical details that need to be included so that this experiment can be reproduced and so that confounding effects can be excluded. For example, this type of experiment is expected to result in mosaic deletion with many cells being spared of the NUA2 deletion. What is the level of deletion on the tissue level? I'm afraid the mRNA staining of sections is not terribly clear. Is there an antibody that can be used for IHC? Is there a genotyping assay to quantitate this deletion? Is there a non-targeting sgRNA control that could be used to rule out non-specific effects of the guide strand targeting and Cas9 expression? Can the sequences be provided?

2. Also, the Cas9 targeted cell lines in Fig. 3 is a nice experiment, but again, there are many technical details that are not described. Are these independent single clones of cells that were transfected or are these pools that were targeted? This makes a big difference in interpretation since a clone would expect to have a strong effect, while pooled cells would not be expected to have a strong effect given the escapers from NUA2 deletion. In Fig 3b, will NUA2 sgRNA also target the rescue vectors? What is the sequence of these vectors?

3. The evidence for the positive feedback loop established through actomyosin tension is weak. I don't think the NLS expt is useful because the fact that more YAP goes to the nucleus would make the rescue of proliferation obvious. How is the Yap translocation to nucleus enhanced when actomyosin is activated?

4. It would be interesting to know if the tumors with NUA2 amplification are Yap low, as would be predicted if this is a central node downstream of Yap/TEAD. Can expression data or TCGA data be provided?

Minor issues:

1. The paragraph starting on Line 187 is an odd place for an intro about NUA2, this should be placed earlier, when the gene is first introduced.

2. This statement and the accompanying discussion should be cited: "While inhibitors for Mst1/2 kinases have been reported."

Reviewers' comments:

Reviewer #1, Expertise: Tyrosine Kinases (Remarks to the Author):

Here the authors identified NUA2, a member of the AMPKR family of kinases, as a YAP target gene in murine hepatocytes, engineered to inducibly express activated YAP S127A, based on an increased level of TEAD4 transcription factor binding to a regulatory element downstream of the Nuak2 gene. Analysis of a series of human cancer lines with activated YAP also showed YAP/TEAD binding to the NUA2 gene, and knockdown of YAP reduced NUA2 expression. Consistently, the levels of NUA2 and YAP RNAs were positively correlated in human HCC samples. AAV-mediated expression of a Nuak2 sgRNA in the liver of Cre-dependent Cas9 mice bearing a Cre-dependent, Dox-inducible human YAP-S127A allele greatly reduced the hepatomegaly and hepatocyte proliferation that is normally induced in these mice by Dox treatment, concomitant with reduced Nuak2 expression and phosphorylation of MYPT1, a known NUA2 substrate that acts a regulatory subunit of PP1 to target phospho-MLC. NUA2 phosphorylation of MYPT1 reduces MYTP1/PP1 phospho-MLC phosphatase activity. When AAV-Cre was administered at a low dose to induce YAP-S127A in a small number of hepatocytes, Nuak2 sgRNA-mediated depletion of Nuak2 reduced the number and size of liver tumors. NUA2 depletion in HCC cells with high YAP levels, such as human HuCCT-1 cells, reduced proliferation of these cells, the expression of YAP/TEAD target genes and MYPT1 phosphorylation. These phenotypes were reversed by expression of WT but not kinase-dead NUA2. Loss of NUA2 resulted in decreased phosphorylation of MYPT1, which in turn increased dephosphorylation of phospho-MLC by MYPT1/PP1. The resultant decrease in MLC phosphorylation led to disassembly of actin fibers, which in turn reduced YAP expression that is known to be stimulated via assembly of the actin cytoskeleton. The same effects of Nuak2 depletion were observed in mouse livers, where there were reduced junctional actin fibers and nuclear YAP. Expression of nuclearly targeted NLS-YAP-S127A partly reversed the decreased proliferation of HuCCT-1 cells observed with NUA2 knockdown. Finally, they showed that HTH-02-006, a semi-specific inhibitor of NUA2 kinase, selectively inhibited proliferation of cancer cell lines with high YAP, and that HTH-02-006 inhibition of NUA2 suppressed YAP-driven hepatomegaly and tumorigenesis in the Dox-inducible YAP-S127A mouse model.

This is well conducted study convincingly demonstrating that the NUA2 protein kinase, a member of the AMPKR family of kinases, is a target gene for the Hippo growth regulatory pathway, whose role is to inhibit proliferation and growth by phosphorylation of the YAP transcriptional co-activator protein to retain it in the cytoplasm and prevent it from activating expression of proliferation-related genes. Their data show that NUA2 is a direct target for YAP/TEAD, and that disruption of YAP in tumor cells with high YAP-expression decreases proliferation in culture, and YAP-S127A-driven hepatocellular carcinoma in mice. The loss of NUA2 kinase activity leads to increased MYPT1/PP1 phosphatase activity, which reduces MLC phosphorylation and decreases actin polymerization, leading in turn to loss of a feedback loop that would normally trigger further YAP expression.

One major question raised by these studies is whether MYPT1 is the most important or only substrate for NUA2 relevant to the cell proliferation and tumorigenesis phenotypes, and there is no discussion of this issue. The alteration in the actin cytoskeleton when NUA2 expression is reduced is consistent with NUA2-mediated MYPT1 phosphorylation playing a role, but there certainly could be other NUA2 substrates that are also important for tumorigenesis.

Response: We thank the reviewer for these insightful comments. As suggested, we

now not only provide new revised data showing that NUA2 exerts its oncogenic roles at least partially through MYPT1 by rescuing the effect of NUA2 loss via MYPT1 knockdown (revised Supplementary Fig. 4e and f), but also discuss other oncogenic proteins regulated by NUA2 in the revised manuscript. As we are primarily focusing on the YAP-NUA2 axis in liver cancer development in this study, and additional in-depth research might be required to fully identify more NUA2 substrates in liver tumorigenesis, we sincerely hope the reviewer will concur that these determinations warrant a separate study and manuscript.

Points: 1. Figure 2d: While the increase in MYPT1 phosphorylation was strong, the increase in Nuak2 protein was modest. Presumably each lane represents a different mouse liver, and this should be indicated in the figure legend.

Response: We believe that the difference in sensitivity by each antibody make it difficult to quantitatively compare fold inductions of NUA2 and phospho-S445 MYPT1 by Western blot. To address the reviewers concern and further demonstrate that YAP induces NUA2 expression, we performed qPCR to examine *NUA2* mRNA expression levels. Consistent with the induction of MYPT1 phosphorylation, it shows that YAP overexpression dramatically increased *NUA2* expression and *NUA2* KO or YAP KO abrogated this phenomenon. The data is now shown in revised Supplementary Fig. 2b.

We thank the reviewer for pointing out our omission, and have corrected all our figure legends to state that each lane represents a different mouse liver.

2. Figure 2: There are many phosphorylation sites mapped in MYPT1, and the authors need to indicate in the Results that have monitored phosphorylation of S445, which is a characterized NUA2 site.

Response: We apologize for the inexact description. We now explicitly state the analyzed NUA2 phosphorylation site of MYPT1 (S445) in the revised manuscript.

3. Figure 3: The authors should explain why Cas9 does not target the NUA2 sgRNAs to act on the co-transfected WT and KD NUA2 construct DNAs.

Response: We thank the reviewer to point this out and apologize for the unclear description. In our experiment, we mutated the PAM sequence of the sgRNA in Nuak2 WT and KD construct DNAs. Thus, the transfected construct DNA would not be targeted by Cas9. We now better describe this information in the revised manuscript.

4. Figure 4a/b: There was a partial reduction of nuclear YAP upon NUA2 depletion (was this quantified and confirmed by cell fractionation?), but the mechanism is not immediately obvious. The authors imply that this was due to the reorganization of the actin cytoskeleton, but was there also decreased phosphorylation of S127 and 14-3-3 binding (this should be checked) or is another mechanisms involved.

Response: We thank the reviewer for these critical and constructive suggestions. As suggested, the quantified data of YAP nuclear localization is now included in revised Figure 4d. For quantification of the YAP distribution, the nuclear area was defined using DAPI image and delineated using Image J software. The fluorescence in the nuclear area was calculated as the percentage of total cell fluorescence by Image J software. Moreover, as suggested, cell fractionation analyses were performed which clearly showed that NUA2 depletion decreases YAP nuclear localization and increases cytosolic YAP levels (revised Fig. 4e). Regarding the underlying mechanisms by which NUA2 regulates YAP subcellular localization, we now demonstrate YAP S127 phosphorylation and 14-3-3 binding of YAP by immunoprecipitation upon NUA2 knockout and have included the data in the manuscript (revised Supplementary Fig. 4b and c). Indeed, YAP S127 phosphorylation and YAP-14-3-3 binding was increased after NUA2 knockdown

suggesting that YAP cytosolic occurs through phosphorylation-mediated 14-3-3 sequestration.

5. Figure 5: Does HTH-02-006 treatment of hepatoma lines reduce YAP protein expression, e.g. by blocking feedback, or decrease NUA2 levels (some inhibitors lead to decreased kinase protein stability)? As the authors suggest, the inhibitory effect of HTH-02-006 treatment on xenograft tumor growth could primarily be due to inhibition of NUA2 activity, but given the possible contribution of off-target effects, what is really needed is re-expression of an HTH-02-006-resistant mutant form of NUA2 in these cells to demonstrate that tumor growth is now unaffected by HTH-02-006 treatment.

Response: As the reviewer suggested, we tested whether HTH-02-006 affects the expression levels of YAP and NUA2. As shown below, we found that HTH-02-006 treatment moderately reduced YAP expression, but showed no effects on NUA2 expression. Moreover, to further demonstrate the specificity of HTH-02-006, we generated a cell line with an A236T mutation in NUA2 endogenous alleles by CRISPR, which is predicted to have lower binding affinity to HTH-02-006. As shown in the Revised Fig. 5e, HTH-02-006 treatment robustly reduced the phosphorylation of MYPT1 in wild-type cells, but not in the *NUAK2^{A236T/236T}* mutant, suggesting that the *NUAK2^{A236T/236T}* mutant could be considered a drug resistant mutant. By comparing this mutant with its wild-type counterpart, *NUAK2^{A236T/236T}* mutant cells showed some resistance to HTH-02-006 treatment (revised Fig. 5f), indicating that the effect of HTH-02-006 treatment on cell growth inhibition is at least partially through NUA2 inhibition.

Reviewer #2, Expertise: Hippo/YAP pathway (Remarks to the Author):

Yuan et al studied the mechanism of YAP in hepatomegaly and tumorigenesis. The authors discovered that NUA2 as a direct downstream target gene of YAP. Importantly, knockdown of NUA2 significantly blocked the hepatomegaly and liver tumors that were induced by active YAP expression. The authors also showed that NUA2 formed a positive feedback loop to activate YAP. Furthermore, they identified NUA2 inhibitors, though not very specific, and showed that pharmacological inhibition of NUA2 could suppress cell growth of YAP high cells as well as liver tumors induced by active YAP expression.

This study identifies a critical downstream target gene, NUA2, in mediating the physiological function of YAP. Moreover, their data suggests that NUA2 could be therapeutic target for YAP-driven cancers. Therefore, the study is interesting and significant for both basic research to understand the mechanism of YAP in cell growth regulation and potential targets for cancer drug development.

Response: We thank the reviewer for recognizing the novelty of our manuscript and its significant impact on both the basic study of Hippo-YAP signaling and the field of cancer drug development. We also thank the reviewer for the constructive criticisms, which we feel have helped us further strengthen our manuscript.

One major remaining issue is whether the NUA2 and its inhibitor HTH-02-006 selectively inhibit growth of YAP dependent cells. Though the authors have tried to address this issue in Fig.5d, but the data is rather weak. The authors test NUA2 knockdown and HTH-02-006 against cancer cells that the Hippo pathway are mutated, such as mutation in NF2 or LATS, to show the specific role of NUA2 in YAP high cell. Alternatively, the authors can use wild type and LATS knockout MEF to demonstrate whether NUA2 knockdown or HTH-0-006 is more sensitive to YAP high cells.

Response: We appreciate the reviewer's constructive comments. As suggested by the reviewer, paired NF2 depleted/mutated cell lines were utilized to test their sensitivity to NUA2 knockdown, as evaluated by cell proliferation assays. In comparison with WT cells, NF2 depleted/mutated cells displayed higher sensitivity to NUA2 knockdown, suggesting the crucial role of NUA2 in YAP-driven tumors. These data are now included in **revised Fig. 3e and 3f**.

Fig.3b, 3d. The NUA2 knockdown efficiency, such as Western of NUA2 protein, should be included. NUA2 is a YAP target gene, but its mRNA is very low in the HuCCT-1 cells. Do the authors have explanation for this?

Response: We now have performed Western Blotting analysis of NUA2 to confirm the knockdown efficiency and included all the data in the **revised Supplementary Fig. 3c, 3g, and 3h**. Moreover, we apologize if our description of the data was unclear. As shown in Figure 3c, Nuak2 mRNA expression in HuCCT1 is much higher than that in SNU398 and HepG2 cells. Accordingly, high YAP-expressing HuCCT-1 cells exhibited higher Nuak2 expression.

Fig.4a. Again, Western for NUA2 should be presented to show the knockdown efficiency.

Response: We have included a Western Blotting analysis of Nuak2 in the **revised Fig. 4a**.

Reviewer #3, Expertise: Liver cancer (Remarks to the Author):

The authors identified an AMPK-related kinase, NUA2, as a potential transcriptional target of the Hippo-YAP signaling pathway. The authors note that this is an important endeavor because the Hippo pathway is difficult to access therapeutically and that there are not that many bona fide direct targets that are amenable to chemical inhibition. They convincingly show that YAP activates NUA2 through TEAD-DNA-binding. NUA2 expression is upregulated in liver cancer cells and is correlated with YAP expression. Mechanistically, NUA2 acts on Myosin phosphatase target subunit (MYPT1), increasing MLC phosphorylation and triggering the assembly of actin fibers. They propose that this forms a positive feedback loop to auto-amplify YAP activity through increased actomyosin tension. Targeting NUA2 by HTH-02-006 attenuates cancer cell line proliferation and progression.

The work is clear and solid, the models are varied and robust, and ultimately I am convinced that this is a good target of Yap and worthy of therapeutic intervention. There are some things that can be clarified or supplemented so that the work can be better interpreted and repeated:

Response: We thank the reviewer for recognizing the significant impact of this study. We also appreciated the reviewer for raising the constructive comments to help us further strengthen our manuscript. Below please find the point-by-point response to the reviewer's critiques

1. The *in vivo* AAV-Cas9 experiment to delete NUA2 is very elegant (fig. 2), however, there are many technical details that need to be included so that this experiment can be reproduced and so that confounding effects can be excluded. For example, this type of experiment is expected to result in mosaic deletion with many cells being spared of the NUA2 deletion. What is the level of deletion on the tissue level? I'm afraid the mRNA staining of sections is not terribly clear. Is there an antibody that can be used for IHC? Is there a genotyping assay to quantitate this deletion? Is there a non-targeting sgRNA control that could be used to rule out non-specific effects of the guide strand targeting and Cas9 expression? Can the sequences be provided?

Response: We appreciate the recognition of the technical accomplishment of the *in vivo* experiments and apologize to the Reviewer for any confusion caused by our unclear description. We have now improved the clarity of our figures and legend texts in our revised manuscript. First, to quantitate the knockout efficiency, fragment assay was performed and has revealed approximately 70% knockout efficiency of the NUA2, as shown in revised Supplementary Fig. 2a. Second, unfortunately, to date, there is no specific NUA2 antibody that can be used for successful immunostaining, and several attempts with different antibodies have failed. We therefore performed an RNA in situ hybridization approach as a surrogate to stain *NUA2* mRNA in liver sections. The data is now included in the revised Fig. 2e.

Lastly, we apologize that we didn't include scramble ctrl sgRNA, but just used empty vector as control, in our AAV-virus system. However, in addition to the AAV system, we established a transposon system to integrate sgRNAs into the genome of hepatocytes by hydrodynamic tail vein injection. In this delivery system, we included two scramble sgRNAs and observed a similar phenotype as with the AAV system. As shown below, compared to scramble control, only Nuak2 depletion by sgRNA for Nuak2 in this system abolished YAP-driven clonal expansion and the ctrl sgRNA experimental livers showed identical clonal expansions as in the empty AAV experiment. We believe this provides sufficient evidence to rule out un-specific effects of the guide strand targeting and Cas9 expression.

The sequences of scramble sgRNAs are described below:

5'- ATGTTGCAGTTCGGCTCGAT-3' and 5'- ACGTGTAAGGCGAACGCCTT-3'

IHC analysis for YAP

2. Also, the Cas9 targeted cell lines in Fig. 3 is a nice experiment, but again, there are many technical details that are not described. Are these independent single clones of cells that were transfected or are these pools that were targeted? This makes a big difference in interpretation since a clone would expect to have a strong effect, while pooled cells would not be expected to have a strong effect given the escapers from NUA2 deletion. In Fig 3b, will NUA2 sgRNA also target the rescue

vectors? What is the sequence of these vectors?

Response: The figure legends have now been revised to describe the performed experiments in a detailed manner.

To answer the reviewer's specific questions here: In Fig. 3a, the transfected cells are pooled. In Fig. 3b, we mutated the PAM seq of sgRNA in NUA2 WT and KD construct DNAs, so that the rescue construct DNA would not be targeted by Cas9. The sgRNAs were cloned into the pLentiCRISPR plasmid and NUA2 cDNAs were cloned into pRK5-Flag expression plasmid. This has now also been described in detail in the materials and methods section.

3. The evidence for the positive feedback loop established through actomyosin tension is weak. I don't think the NLS expt is useful because the fact that more YAP goes to the nucleus would make the rescue of proliferation obvious. How is the Yap translocation to nucleus enhanced when actomyosin is activated?

Response: We acknowledge the comment by the reviewer and indeed agree that the use of NLS-YAP might not totally explain the positive feedback loop. To achieve this goal, we have performed another experiment to demonstrate this positive feedback loop. As shown in the revised Supplemental Fig. 4e and f, depletion of MYPT1 in NUA2-knockdown cells partially rescued not only cell growth but also YAP transcriptional activity mediated by NUA2 depletion. These data suggest that NUA2 regulates cell proliferation and amplifies YAP activity at least in part through MYPT1, which is known to suppress actomyosin tension.

How YAP nuclear localization is mediated by the activation of actomyosin is still an open question in the field. We sincerely hope the reviewer will concur that these determinations warrant a separate study and manuscript, as we are primarily focusing on the YAP-NUA2 axis in liver cancer development in this study and additional in-depth research might be required to fully decipher the underlying mechanisms by which actomyosin activation regulates YAP nuclear localization. However, we tried to test whether NUA2 depletion affect YAP S127 phosphorylation and 14-3-3 binding, which is a well-known mechanism, regulated YAP localization. In revised Supplementary Fig. 4b and c, we showed that YAP S127 phosphorylation and YAP-14-3-3 binding was increased after *NUA2* knockdown suggesting that YAP cytosolic occurs through phosphorylation-mediated 14-3-3 sequestration.

4. It would be interesting to know if the tumors with NUA2 amplification are Yap low, as would be predicted if this is a central node downstream of Yap/TEAD. Can expression data or TCGA data be provided?

Response: We apologize for the unclear descriptions that have confused the reviewer. Our working model is a new auto-regulatory positive feedback loop to maximize YAP activity through NUA2. In fact, the effector, which regulates cell proliferation and tumorigenesis is still YAP, but not NUA2. Therefore, we respectfully disagree with the reviewer that NUA2 amplification should be associated with YAP low levels in human specimens. As shown below, NUA2 amplification did not show any correlation with YAP levels from TCGA liver cancer dataset.

Minor issues:

1. The paragraph starting on Line 187 is an odd place for an intro about NUAK, this should be placed earlier, when the gene is first introduced.

Response: We thanks for the reviewer's suggestion. We have now replaced this section earlier in the revised manuscript.

2. This statement and the accompanying discussion should be cited: "While inhibitors for Mst1/2 kinases have been reported."

Response: We apologize to the reviewer for our oversight. We have cited the reference accordingly.

Reviewers' Comments:

Reviewer #1:

Remarks to the Author:

X

Reviewer #3:

Remarks to the Author:

Most of my questions have been addressed, but some issues remain. Supp Fig. 2a does not quite answer the question about Nuak deletion. This shows that 68% of the Nuak2 alleles are deleted. That I understand. But how many cells are knocked out? for example, 50% allele deletion could mean that 100% of cells are heterozygous and none are KOs. Assuming that there is a 68% editing efficiency, then you would assume that 44% KO rate based on 68% x 68% deletion rate for each allele. If that were the case, then the images in Fig 2e do not make sense. The WT image shows no Red signal for the mRNA, yet the middle images show robust signal. And the right hand image shows essentially complete deletion. This does not provide greater confidence in the technical aspects of the knockout quantification. Cas9 experiments in vivo need to have very thorough characterization because of these mosaicism issues. Also, empty vectors really are not a suitable control for sgRNAs.

Reviewer #4:

Remarks to the Author:

This is an important and interesting manuscript identifying NUA2 as an important direct transcriptional target of YAP and a crucial target in the development of liver cancer. The first 3 reviewers did an excellent job of critiquing the original manuscript and made several good suggestions for improvement. The response of the authors was very good, providing additional data or explanations when appropriate. I agree with reviewer 3 that the evidence for the positive feedback loop established through actomyosin tension is weak, but this is due more to a deficiency in the hippo field than the authors' experimental limitations. The notion of actomyosin regulation of YAP is widely held, but since there is no clear molecular pathway known for this, it is difficult to test directly (actomyosin does so many different things in the cell). Nonetheless, I agree with the reviewer that this finding, although potentially interesting, is not central to their discovery of the YAP-NUA2 link.

Reviewers' comments:

Reviewer #1, Expertise: protein tyrosine kinases (Remarks to the Author):

X

Reviewer #3, Expertise: liver cancer (Remarks to the Author):

Most of my questions have been addressed, but some issues remain. Supp Fig. 2a does not quite answer the question about Nuak deletion. This shows that 68% of the Nuak2 alleles are deleted. That I understand. But how many cells are knocked out? For example, 50% allele deletion could mean that 100% of cells are heterozygous and none are KOs. Assuming that there is a 68% editing efficiency, then you would assume that 44% KO rate based on 68% x 68% deletion rate for each allele. If that were the case, then the images in Fig 2e do not make sense. The WT image shows no Red signal for the mRNA, yet the middle images show robust signal. And the right hand image shows essentially complete deletion. This does not provide greater confidence in the technical aspects of the knockout quantification. Cas9 experiments in vivo need to have very thorough characterization because of these mosaicism issues. Also, empty vectors really are not a suitable control for sgRNAs.

Response:

We thank the reviewer for his/her comments and therefore generated AAV containing two non-targeting sgRNAs with Cre as controls and have included this piece of data below and in Supplementary Fig. 2a. Compared with AAV-Cre, TetO-YAP:Cas9 mice infected with AAV-Cre non-targeting sgRNA do not show any difference. Most importantly, AAV-Cre-sgRNA-Nuak2 demonstrated a substantial and highly significant reduction in liver overgrowth. This data not only further confirms the role of Nuak2 in YAP-driven hepatomegaly but also rule out non-specific effects of the guide strand targeting and Cas9 expression.

Gross morphology of the livers of TetO-YAP:Cas9 transgenic mice infected with indicated AAV virus and placed on Dox for 2 weeks. Liver/body weight ratio of mice mentioned before was plotted. Data are presented as mean \pm SD; n=3.

The sequences of non-targeting sgRNAs are described below:

5'- ATGTTGCAGTTCGGCTCGAT-3' and 5'- ACGTGTAAGGCGAACGCCTT-3'

Regarding the knockout efficiency issues raised by the reviewer, we recognize his/her concerns, and would like to address them as below.

1) In the mentioned experiment showing 68% deletion on a genomic DNA level, it has to be considered that the DNA was isolated from a piece of liver tissue. While hepatocytes are the main cell type in the liver, there is still a significant amount of other non-parenchymous cells present such as endothelial cells, biliary cells and others, which are not infected by the AAV2/8 and therefore will not delete *Nuak2*. This leads to an underestimation of the *Nuak2* deletion efficiency in hepatocytes and make the data not entirely comparable to the histological analysis where the focus was specifically on hepatocytes.

We made a remark in the revised result section on the limitations of comparability.

2) We completely agree with the reviewer that by bulk genomic analysis we cannot quantitatively assess the *Nuak2* deletion allele frequency per cell and thus the expected *Nuak2* KO mosaicism. To that purpose, we had included RNAscope of liver tissue sections to better assess KO efficiency per cell (Figure 2e). Upon the reviewers remarks we revisited the images and realized that the immunofluorescence intensity was dim on these images, making proper comparisons difficult. As shown below, after a general increase of the intensity in all images we believe it should be clearer now that not all hepatocytes in the Yap overexpression/sg*Nuak2* sample (red dot) are completely knocked out. Furthermore, we include another field to show very similar results. In that field, it clearly displays that there are some regions with higher *Nuak2* mRNA expression (arrow).

Nevertheless, even heterozygous *Nuak2* deletion leads to some reduction of *Nuak2* expression, thus in turn abrogating YAP activity and liver tumorigenesis, which is completely in line with our experimental data and the proposed working model.

Reviewer #4, Reviewer replacement for Reivewer #3, Expertise: Hippo/YAP pathway(Remarks to the Author):

This is an important and interesting manuscript identifying NUAK2 as an important direct transcriptional target of YAP and a crucial target in the development of liver cancer. The first 3 reviewers did an excellent job of critiquing the original manuscript and made several good suggestions for improvement. The response of the authors was very good, providing additional data or explanations when appropriate. I agree with reviewer 3 that the evidence for the positive feedback loop established through actomyosin tension is weak, but this is due more to a deficiency in the hippo field than the authors experimental limitations. The notion of actomyosin regulation of YAP is widely held, but since there is no clear molecular pathway known for this, it is difficult to test directly (actomyosin does so many different things in the cell). Nonetheless, I agree with the reviewer that this finding, although potentially interesting, is not central to their discovery of the YAP-NUAK2 link.

Response: We thank the reviewer for recognizing the novelty and the significant impact of this study on both the cell signaling and the cancer biology fields. We also appreciated the reviewer for agreeing that we are primarily focusing on the YAP-NUAK2 axis in liver cancer development in this study and additional in-depth research might be required to fully decipher the underlying mechanisms by which actomyosin activation regulates YAP nuclear localization.